# Enriched Finite Element Method Based on Interpolation Covers for Structural Dynamics Analysis

Qiyuan Gu [1,†] , Hongju Han [2,†], Guo Zhou [1], Fei Wu [3,*] , Zegang Ju [1], Man Hu [1], Daliang Chen [4] and Yaodong Hao [4]

1   College of Engineering and Technology, Southwest University, Chongqing 400715, China; 13101103285@163.com (Q.G.)
2   Guizhou Road and Bridge Group Co., Ltd., Guiyang 550001, China; hanhongjuhan@163.com
3   College of Mechanical and Vehicle Engineering, Chongqing University, Chongqing 400044, China
4   China Automotive Technology Research Center Co., Ltd., Tianjin 300300, China
*   Correspondence: wufeifrank@cqu.edu.cn
†   These authors contributed equally to this work.

**Abstract:** This work proposes a novel enriched finite element method (E-FEM) for structural dynamics analysis. We developed the enriched 3-node triangular and 4-node tetrahedral displacement-based elements (T-elements). The standard linear shape functions of these T-elements were enriched using interpolation cover functions over each patch of elements. We also introduced and compared different orders of cover functions; higher-order functions obtained higher computational performance. Subsequently, the forced and free vibration analyses were performed on various typical numerical examples. The proposed enriched finite element method generated more precise numerical results and ensured faster convergence than the original linear elements.

**Keywords:** low-order linear element; interpolation cover function; enriched finite element method (E-FEM); forced and free vibration analysis





## 1. Introduction

The recent use of FEM (finite element method) [1] has intensified in various engineering fields. Low-order three-node triangular elements [2] and four-node tetrahedral displacement-based elements [3] are commonly used in FEM due to its benefits of simple construction in complex structures. However, conventional low-order T-meshes still have defects in computational precision and sensitivity to mesh distortion. The "over-stiffness" properties of these T-meshes often yield poor computational outcomes (higher nature frequency) in the structural dynamics analysis. Significant research efforts are essential to improve the performance of these T-meshes.

Liu et al. [4] proposed a well-known strategy that incorporated gradient smoothing technology into the FEM. Based on alternative ways of constructing smoothing domains, the "over-stiffness" stiffness matrix of conventional FEM could be appropriately softened to a certain extent. Therefore, various S-FEMs [5–9] (Smoothed FEM) are proposed, including alpha FEM (α-FEM), beta FEM (β-FEM), face-based smoothed FEM (FS-FEM), node-based smoothed FEM (NS-FEM), edge-based smoothed FEM (ES-FEM), cell-based smoothed FEM (CS-FEM) [10], among other variations. In contrast with corresponding FEM, S-FEM yields more precise outcomes with higher convergence speed without increasing the relative computational cost [11–14]. The NS-FEM harbors an excessive softening effect on the system stiffness, resulting in unstable false shapes in structural dynamics analysis [11,12]. Unlike the "overly-stiff" FEM and the "overly-soft" NS-FEM model, the ES-FEM model has a closely exact stiffness. Besides, the ES-FEM has good mesh distortion adaptability [13–17]. For this reason, the ES-FEM is selected for comparison with the proposed approaches.

Another technique, the Enriched-FEM, was recently developed and used in various fields, including crack propagation, wave propagation, acoustic problems, multi-physical coupling, and solid mechanics. Bathe [18–22] and his colleagues first proposed this numerical approach to resolve issues in static analysis of solid mechanics and wave propagation. Wu [23] et al. adopted the E-FEM to efficiently minimize dispersion error at high frequencies and solve acoustic issues by applying the interpolation cover functions to improve the convergence rate and precision of solutions. The pertinent numerical findings indicate that the E-FEM may significantly minimize numerical dispersion error in wave propagation and considerably manufacture stable and more precise numerical solutions. Zhou [24] (2022) et al. utilized the E-FEM to analyze the dynamics of Magneto-electro-elastic (MEE) smart structures. The E-FEM outcomes used to examine the multi-physical coupling are consistent with the analytical analysis.

Furthermore, a variety of typical MEE-based specific examples have been used to reveal how efficiently the current E-FEM handles multi-physical coupling issues, unlike the conventional FEM. Li [25] and colleagues employed the supplementary interpolation cover functions, which are constructed using appropriate polynomial bases, to improve the performances of the traditional FEM at the same time handling 2D dynamic scenarios. As a result, the gradient field of the issue under examination can be more precisely described, and the initial linear approximation space of the conventional FEM can be considerably enriched [18–27]. In summary, E-FEM has extensive application prospects.

Herein, we propose a novel enriched finite element method with 3-node triangular and 4-node tetrahedral displacement-based elements for structural dynamic analysis. We also introduce and compare different orders of the cover functions using the standard FEM and recently developed ES-FEM. The enriched interpolation cover functions are constructed rather flexibly using different orders of the base functions. Numerical examples demonstrate that the proposed method provides more accurate numerical results and ensures faster convergence, unlike the original linear elements.

The rest of the article is organized as follows. The fundamental FEM theoretical formula and the E-FEM formula for structural dynamics analysis is thoroughly introduced in Section 2. The performance of the E-FEM is assessed using several numerical examples in Sections 3 and 4. The conclusions are given in Section 5.

## 2. Formula of the E-FEM

Imagine that a typical finite element mesh has been created to solve a physical issue. The type of element and the mesh employed determine the precision of the sought-after solution. In order to improve the finite element procedure, we use the numerical manifold method's earliest strategy and create tiny, overlapping sub-domains [28,29], where the finite elements in the supplied finite element mesh are the common parts of the sub-domains. Due to the interpolation covers utilized to cover each sub-domain, a higher-order interpolation of the desired solution is made feasible, increasing the accuracy of the results. Since our main goal is to enhance the performance of discretization utilizing these elements, we will now consider the 2D and 3D analytical situations using, respectively, the 3-node and 4-node low-order elements. Since they are trustworthy for both linear and nonlinear solutions, we use the traditional low-order finite elements; their fundamental flaw is the lack of precision in their results.

### 2.1. Theory of the E-FEM

The relevant issue domain is discretized using the common triangular elements with $N$ nodes. And the interpolation for a scalar field function $d$ using the conventional FE approximation takes the following form.

$$d = \sum_{i=1}^{N} N_i d_i = \mathbf{N}\mathbf{d} \tag{1}$$

where $N_i$ denotes the piece-wise linear shape function and $d_i$ stands for the nodal field variable (see Figure 1a).

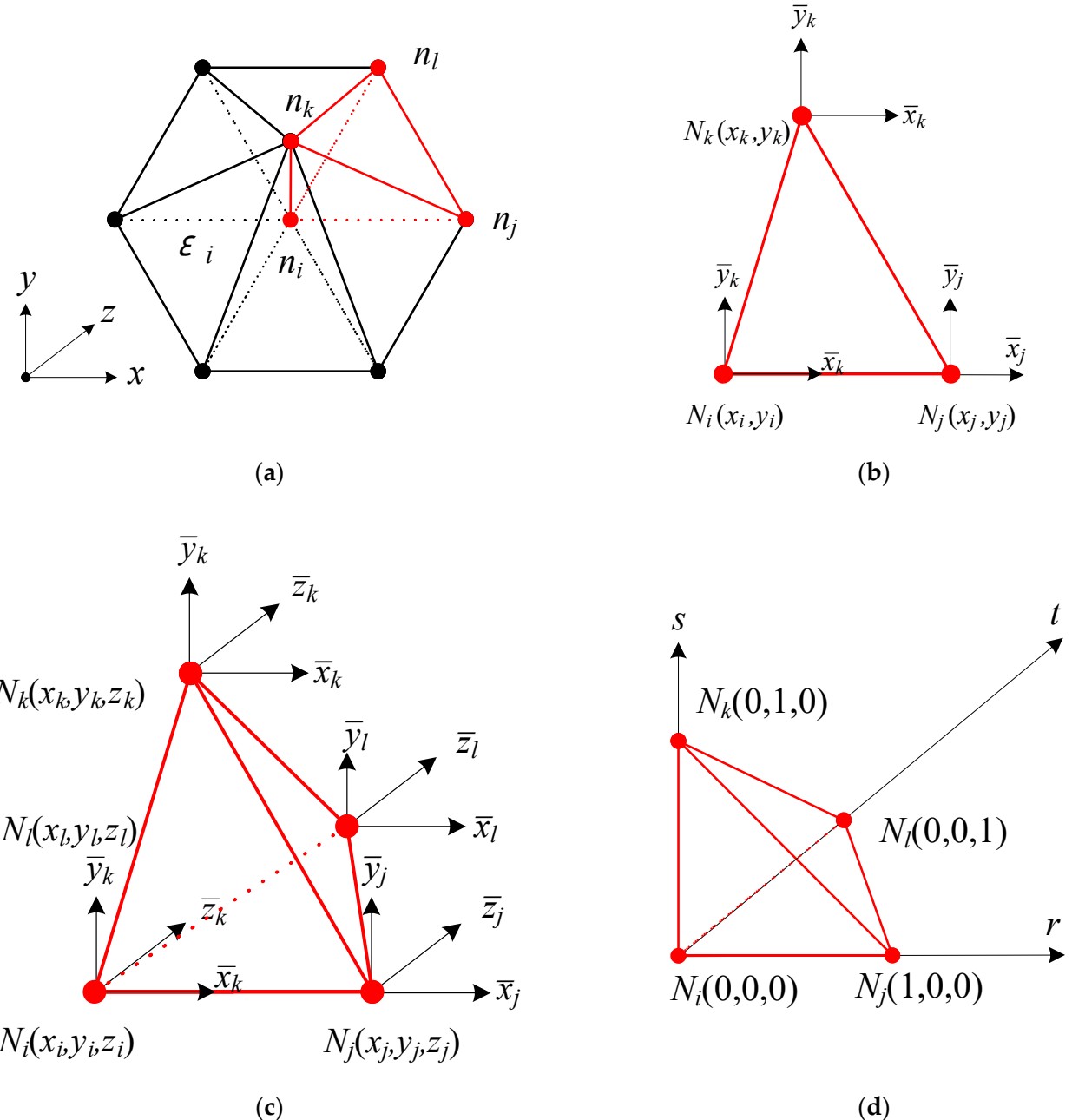

**Figure 1.** The enriched cover interpolation sub-domains that employ a standard triangular mesh (**a**,**b**) and tetrahedral mesh (**c**,**d**) for E-FEM: (**a**) The cover region ($\varepsilon_i$) for enriched interpolation covers and the typical linear interpolation functions (**b**) Local coordinates and physical coordinates of triangular element nodes; (**c**) Displacement of tetrahedral element node after introducing basis function; (**d**) The natural coordinate system of the tetrahedral element.

The traditional finite element (FE) mesh is still used in the current E-FEM. As seen in Figure 1a, the global elements connected to the node $i$ which is designated as the cover region $\varepsilon_i$ constitutes the support domain of the conventional linear nodal interpolation function in the standard FE interpolation. The following equation for the taken-in field variable $d$ at node $i$ improves the FE interpolation.

$$E_i^q[d] = \mathbf{L}\begin{bmatrix} d_i & a_{i1} & a_{i2} & a_{i3} & \cdots \end{bmatrix}^T \tag{2}$$

in which **L** refers to the degree $q$ polynomial bases that are applied over the cover region $\varepsilon_i$, $d_i$ denotes the conventional nodal variable, and $\begin{bmatrix} a_{i1} & a_{i2} & a_{i3} & \cdots \end{bmatrix}^T$ is the extra unknowns connected to the interpolation cover functions.

$$\mathbf{L} = \begin{bmatrix} 1 & \overline{x}_i & \overline{y}_i & \overline{x}_i^2 & \overline{x}_i\overline{y}_i & \overline{y}_i^2 & \cdots & \overline{y}_i^q \end{bmatrix} \tag{3}$$

where $\overline{x}_i = x - x_i$ and $\overline{y}_i = y - y_i$ are the relative coordinate values $(\overline{x}_i, \overline{y}_i)$ calculated from node $i$. (See Figure 1b)

Utilizing interpolation cover functions, we assign a group of full polynomial bases to each cover region in order to enrich the traditional FE interpolation for the solution of the variable $d$. Equation (2) can also be rewritten to provide a clear comparison with the common FE interpolation.

$$E_i^q[d] = d_i + \begin{bmatrix} \overline{x}_i & \overline{y}_i & \overline{x}_i^2 & \overline{x}_i\overline{y}_i & \overline{y}_i^2 & \cdots & \overline{y}_i^q \end{bmatrix} \underbrace{\begin{bmatrix} a_{i1} \\ a_{i2} \\ a_{i3} \\ \vdots \\ a_{iq} \end{bmatrix}}_{\text{additional interpolation cover}} \tag{4}$$

The interpolation scheme of E-FEM, which differs significantly from the current method from the traditional interpolation scheme of FEM, is shown to contain the additional interpolation cover in Equation (4).

Then, the global approximation for the regional variable $d$ can be obtained by

$$TE_i^q[d] = \sum_1^k \left( \sum_{i=1}^3 N_i E_i^q[d] \right) = \sum_1^k \left( \sum_{i=1}^3 N_i d_i + \sum_{i=1}^3 \boldsymbol{H}_i \boldsymbol{a}_i \right) = \sum_1^k \left( \sum_{i=1}^3 N_i d_i + \sum_{i=1}^3 \sum_{j=1}^n H_{i,j} a_{i,j} \right) \tag{5}$$

where $k$ denotes the number of all nodes in $\Omega$, $N_i$ denotes the linear interpolation for the nodes, $i$ and $j$ denote the additional degree of freedom in each node, and **H** is a matrix mixed with a q-order interpolation function.

$$\mathbf{H}_i = N_i \begin{bmatrix} \overline{x}_i & \overline{y}_i & \overline{x}_i^2 & \overline{x}_i\overline{y}_i & \overline{y}_i^2 & \cdots & \overline{y}_i^q \end{bmatrix} \tag{6}$$

The current E-FEM will become the standard FEM, as demonstrated by Equations (2) and (5) if the polynomial bases utilized only have the constant term 1 (i.e., $q = 0$). From this vantage point, it is possible to think of the current interpolation of E-FEM as including additional higher-order terms in addition to the traditional FE interpolation. The higher-order interpolation of solutions can be achieved using this technique. As a result, even with lower-order linear elements, a higher convergence rate and better numerical solution accuracy can be obtained.

In this study, the interpolation cover functions contain first-order polynomial bases $\begin{bmatrix} 1 & x & y \end{bmatrix}$ $(q = 1)$, second-order polynomial bases $\begin{bmatrix} 1 & x & y & x^2 \end{bmatrix}$ $(q = 2)$ and second order polynomial bases $\begin{bmatrix} 1 & x & y & x^2 & xy & y^2 \end{bmatrix}$ $(q = 2)$, and three, four, and six additional DOFs are then introduced for each node for the 2D analysis, respectively. Then, four more DOFs will be inserted for each node for 3D analysis, and the interpolation cover functions will be first-order polynomial bases $\begin{bmatrix} 1 & x & y & z \end{bmatrix}$ $(q = 1)$.

Any order of polynomial bases may be utilized with interpolation cover functions and using high-order polynomial interpolation cover functions can also produce more accurate numerical results. However, the high order interpolation cover functions could increase the number of nodal unknowns and the computational cost.

For the 2D and 3D solid problem, the above-formulated interpolation of the E-FEM approach can be used to acquire the field variables $d$ and $v$ as follows:

$$d_h^k = \sum_{i=1}^{6}(N_i d_i + \mathbf{H}_i \boldsymbol{a}_i^u), v_h^k = \sum_{i=1}^{6}(N_i v_i + \mathbf{H}_i \boldsymbol{a}_i^v) \tag{7}$$

where $\boldsymbol{a}_i^u$ and $\boldsymbol{a}_i^v$ denote the vectors that hold additional unknowns, $u_i$ and $v_i$ denote the normal nodal unknowns as in the standard FEM.

Interpolation is enriched by using the unknowns arranged appropriately in the vector. Then $d_h^k$ and $v_h^k$ become

$$\begin{bmatrix} d_h^k \\ v_h^k \end{bmatrix} = \begin{bmatrix} \mathbf{N} & \mathbf{H} & \mathbf{0} & \mathbf{0} \\ \mathbf{0} & \mathbf{0} & \mathbf{N} & \mathbf{H} \end{bmatrix} \begin{bmatrix} \mathbf{d} \\ \boldsymbol{a}_d \\ \mathbf{v} \\ \boldsymbol{a}_v \end{bmatrix} \tag{8}$$

where $\mathbf{H}$ denotes the supplementary interpolation cover function matrix, $\mathbf{N}$ stands for the conventional FEM interpolation function matrix, $\mathbf{d}$ and $\mathbf{v}$ are the typical nodal displacement vectors, and $\boldsymbol{a}_v$ and $\boldsymbol{a}_d$ stand for the additional unknown solution coefficient vectors.

Applying the common differentiation techniques to the derivatives of the displacement in the interpolation of E-FEM,

$$\begin{bmatrix} d_{h,x}^k \\ d_{h,y}^k \end{bmatrix} = \begin{bmatrix} \mathbf{N}_{,x} & \mathbf{H}_{,x} \\ \mathbf{N}_{,y} & \mathbf{H}_{,y} \end{bmatrix} \begin{bmatrix} \mathbf{d} \\ \boldsymbol{a}_d \end{bmatrix}, \begin{bmatrix} v_{h,x}^k \\ v_{h,y}^k \end{bmatrix} = \begin{bmatrix} \mathbf{N}_{,x} & \mathbf{H}_{,x} \\ \mathbf{N}_{,y} & \mathbf{H}_{,y} \end{bmatrix} \begin{bmatrix} \mathbf{v} \\ \boldsymbol{a}_v \end{bmatrix} \tag{9}$$

where

$$\begin{bmatrix} \mathbf{N}_{,x} & \mathbf{H}_{,x} \\ \mathbf{N}_{,y} & \mathbf{H}_{,y} \end{bmatrix} = \mathbf{J}^{-1} \begin{bmatrix} \mathbf{N}_{,r} & \mathbf{H}_{,r} \\ \mathbf{N}_{,s} & \mathbf{H}_{,s} \end{bmatrix} \tag{10}$$

where the usual FE formulation is used to generate the coordinate transformation between the natural coordinate ($r$, $s$, $t$) and the physical coordinate ($x$, $y$, $z$) (see Figure 1d), and $\mathbf{J}$ is the Jacobian matrix.

From the above formulas, the shape function matrix and the solid mechanical strain matrix $\mathbf{B}$ can be expressed as follows.

$$\mathbf{N}_i = \begin{bmatrix} \mathbf{N} & \mathbf{N}x & \mathbf{N}y & \mathbf{N}x^2 & \mathbf{N}xy & \mathbf{N}y^2 & 0_{1\times3} & 0_{1\times3} & 0_{1\times3} & 0_{1\times3} & 0_{1\times3} & 0_{1\times3} \\ 0_{1\times3} & 0_{1\times3} & 0_{1\times3} & 0_{1\times3} & 0_{1\times3} & 0_{1\times3} & \mathbf{N} & \mathbf{N}x & \mathbf{N}y & \mathbf{N}x^2 & \mathbf{N}xy & \mathbf{N}y^2 \end{bmatrix} \tag{11}$$

$$\mathbf{B} = \nabla \mathbf{N}_i = \begin{bmatrix} \partial \mathbf{N}/\partial x & \partial \mathbf{H}/\partial x & 0_{1\times3} & 0_{1\times15} \\ 0_{1\times3} & 0_{1\times15} & \partial \mathbf{N}/\partial y & \partial \mathbf{H}/\partial y \\ \partial \mathbf{N}/\partial y & \partial \mathbf{H}/\partial y & \partial \mathbf{N}/\partial x & \partial \mathbf{H}/\partial x \end{bmatrix} \tag{12}$$

### 2.2. 3D Structural Element Construction Theory of the E-FEM

A 3D analysis domain $\Omega$ is discretized into several elements (taking the tetrahedral element as an example, the 2D element has less freedom than the 3D element). The basis function $e$ of [1 $x$ $y$ $z$] is first introduced at the node of the tetrahedral element in accordance with the characteristics of node displacement (Note: different types of basic functions can be selected, here only the simplest [1 $x$ $y$ $z$] is taken as an example). The node displacement of the element after introducing the basis function is shown in Figure 1c.

The node displacements are interpolated and described by:

$$\begin{cases} u_i = u_{i1} + xu_{i2} + yu_{i3} + zu_{i4} \\ v_i = v_{i1} + xv_{i2} + yv_{i3} + zv_{i4} \\ w_i = w_{i1} + xw_{i2} + yw_{i3} + zw_{i4} \end{cases} \tag{13}$$

Then the displacement of any point in the element is redescribed:

$$\text{d} = \sum_{i=1}^{N} \mathbf{N}_i \mathbf{d}_i = \mathbf{N}_i \left\{ \begin{array}{c} u_{i1} + x u_{i2} + y u_{i3} + z u_{i4} \\ v_{i1} + x v_{i2} + y v_{i3} + z v_{i4} \\ w_{i1} + x w_{i2} + y w_{i3} + z w_{i4} \end{array} \right\} = \mathbf{N}_{\text{E-FEM}} \mathbf{d}^e_{\text{E-FEM}} \tag{14}$$

where $\mathbf{N}_{\text{E-FEM}}$ denotes the new shape function matrix and $\mathbf{d}^e_{\text{E-FEM}}$ denotes the node displacement vector after rearrangement, which is expressed as:

$$\{ u_{11}\ u_{21}\ u_{31}\ u_{41}\ u_{12}\ u_{22}\ u_{32}\ u_{42}\ u_{13}\ u_{23}\ u_{33}\ u_{43}\ u_{14}\ u_{24}\ u_{34}\ u_{44}\ v_{11}\ v_{21}\ v_{31}\ v_{41}\ v_{12}\ v_{22}\ v_{32}$$
$$v_{42}\ v_{13}\ v_{23}\ v_{33}\ v_{43}\ v_{14}\ v_{24}\ v_{34}\ v_{44}\ w_{11}\ w_{21}\ w_{31}\ w_{41}\ w_{12}\ w_{22}\ w_{32}\ w_{42}\ w_{13}\ w_{23}\ w_{33}\ w_{43}\ w_{14}\ w_{24}\ w_{34}\ w_{44} \}^T$$

The basis function $[1\ x\ y\ z]$ is expressed as $E^T = [e_1\ e_2\ e_3\ e_4]$. It is possible to construct a new shape function matrix $\mathbf{N}_{\text{E-FEM}}$ from the rearranged node displacement vector $\mathbf{d}^e_{\text{E-FEM}}$ as follows:

$$\mathbf{N}_{\text{E-FEM}} = \mathbf{NE} \tag{15}$$

where

$$\mathbf{N} = \begin{bmatrix} N_1 & 0 & 0 & N_2 & 0 & 0 & N_3 & 0 & 0 & N_4 & 0 & 0 \\ 0 & N_1 & 0 & 0 & N_2 & 0 & 0 & N_3 & 0 & 0 & N_4 & 0 \\ 0 & 0 & N_1 & 0 & 0 & N_2 & 0 & 0 & N_3 & 0 & 0 & N_4 \end{bmatrix} \tag{16}$$

$$\mathbf{E} = \begin{bmatrix} E_1 & E_0 & E_0 & E_2 & E_0 & E_0 & E_3 & E_0 & E_0 & E_4 & E_0 & E_0 \\ E_0 & E_1 & E_0 & E_0 & E_2 & E_0 & E_0 & E_3 & E_0 & E_0 & E_4 & E_0 \\ E_0 & E_0 & E_1 & E_0 & E_0 & E_2 & E_0 & E_0 & E_3 & E_0 & E_0 & E_4 \end{bmatrix}^T \tag{17}$$

where

$$E_0 = [0\ 0\ 0\ 0\ 0\ 0\ 0\ 0\ 0\ 0\ 0\ 0\ 0\ 0\ 0\ 0] \tag{18}$$

$$E_1 = [e_1\ 0\ 0\ 0\ e_2\ 0\ 0\ 0\ e_3\ 0\ 0\ 0\ e_4\ 0\ 0\ 0] \tag{19}$$

$$E_2 = [0\ e_1\ 0\ 0\ 0\ e_2\ 0\ 0\ 0\ e_3\ 0\ 0\ 0\ e_4\ 0\ 0] \tag{20}$$

$$E_3 = [0\ 0\ e_1\ 0\ 0\ 0\ e_2\ 0\ 0\ 0\ e_3\ 0\ 0\ 0\ e_4\ 0] \tag{21}$$

$$E_4 = [0\ 0\ 0\ e_1\ 0\ 0\ 0\ e_2\ 0\ 0\ 0\ e_3\ 0\ 0\ 0\ e_4] \tag{22}$$

After obtaining the shape function of the augmented FEM, the geometric matrix of the augmented FEM can be further written out:

$$\mathbf{B} = \mathbf{LN}_{\text{E-FEM}} = \mathbf{LNE} \tag{23}$$

*2.3. Dynamics Controlling Equations for Linear Elastic Solids*

This section comprehensively introduces the discretization system equation and the standard FEM formula of the solid mechanics' problem in the hypothetical bounded domain $\Omega$. The standard Galerkin weak formula is given by:

$$\int_{\Omega} (\bigtriangledown \delta \mathbf{d})^T \mathbf{D} \bigtriangledown \mathbf{d}\, d\Omega = \int_{\Omega} \delta \mathbf{d}^T \left( \widetilde{\mathbf{b}} - \rho \ddot{\mathbf{d}} - c \dot{\mathbf{d}} \right) d\Omega + \int_{\Gamma_{\text{N}}} \delta \mathbf{d}^T t d\Gamma \tag{24}$$

where $\bigtriangledown$ is the differential operator, $\ddot{\mathbf{d}}$, $\dot{\mathbf{d}}$, and $\mathbf{d}$ denotes the acceleration, velocity vectors, and displacement vectors, respectively; the random virtual displacement vector is called $\delta \mathbf{d}$, $\widetilde{\mathbf{b}}$ stands for the body force vector, $\mathbf{D}$ denotes the constant matrix, $\rho$ and $c$ stand for the density and damping coefficients of the materials under consideration, and $\Gamma$ denotes the prescribed traction vector on $\Gamma_{\text{N}}$ which denotes the natural boundary condition.

The matrix version of Equation (24) can be generated using the E-FEM interpolation technique by:

$$\mathbf{M}\ddot{\mathbf{d}} + \mathbf{C}\dot{\mathbf{d}} + \mathbf{K}\mathbf{d} = \mathbf{F} \tag{25}$$

$$\mathbf{M} = \sum_{i=1}^{n_e} \int_{\Omega_i} N_i^T \rho N_i d\Omega \tag{26}$$

$$\mathbf{C} = \sum_{i=1}^{n_e} \int_{\Omega_i} N_i^T c N_i d\Omega \tag{27}$$

$$\mathbf{K} = \sum_{i=1}^{n_e} \int_{\Omega_i} B_i^T D B_i d\Omega \tag{28}$$

$$\mathbf{F} = \sum_{i=1}^{n_e} \int_{\Omega_i} N_i^T \overline{b} d\Omega + \sum_{i=1}^{n_b} \int_{\overline{\tau}} N_i^T t d\tau \tag{29}$$

$$\mathbf{D} = \frac{E}{1-v^2} \begin{bmatrix} 1 & v & 0 \\ v & 1 & 0 \\ 0 & 0 & \frac{1-v}{2} \end{bmatrix} \tag{30}$$

where **M**, **C**, **K**, and **D** denote the matrix of global mass, the damping effects, the global stiffness, and the material parameters, respectively, the global mesh's total element count and the number of elements on the Neumann boundary are indicated by the characters $n_e$ and $n_b$, respectively, **F** denotes the nodal force vector, $B_i$ and $N_i$ denote the strain gradient matrix and the shape function matrix, respectively, $\Omega_i$ stands for element $i$.

*2.4. The Eigenvalue Problem of Free Vibration Analysis*

If the damping effects (**C** = 0) are ignored, it is feasible to rewrite Equation (25) for free vibration analysis by

$$\mathbf{M}\ddot{\mathbf{d}} + \mathbf{K}\mathbf{d} = 0 \tag{31}$$

It is straightforward to determine that Equation (31) has the following basic solution,

$$\mathbf{d} = \overline{\mathbf{d}} \exp(j\omega t) \tag{32}$$

where $\overline{\mathbf{d}}$ denotes the amplitude of displacement distributions, $j = \sqrt{-1}$, and $\omega$ denotes the angular frequency. We may obtain Equation (33) from Equations (31) and (32).

$$\mathbf{K}\overline{\mathbf{d}} - \omega_p{}^2 \mathbf{M}\overline{\mathbf{d}} = 0 \tag{33}$$

The *p*-order natural frequency $\omega_p$ and the corresponding modal shape can be determined by adding up the eigenvectors and eigenvalues of Equation (33). The solution to the common eigenvalue problem is what Equation (33) indicates is the main goal of evaluating free vibration problems.

*2.5. The Dynamic Problem of Forced Vibration Analysis*

The second-order time-dependent dynamic issues, regulated by the matrix equation indicated in Equation (25), should be solved to perform forced vibration analysis [30,31]. Numerous other direct time integration strategies have been established in practice to solve structural dynamic issues. The widely used Newmark approach, an unconditionally stable direct time integration methodology, analyzes dynamic problems, and makes the following assumptions.

$$\begin{cases} {}^{t+\Delta t}\dot{\mathbf{d}} = {}^{t}\dot{\mathbf{d}} + \left[ (1-\beta){}^{t}\ddot{\mathbf{d}} + {}^{t+\Delta t}\ddot{\mathbf{d}}\beta \right] \Delta t \\ {}^{t+\Delta t}\mathbf{d} = {}^{t}\mathbf{d} + {}^{t}\dot{\mathbf{d}}\Delta t + \left[ \left(\frac{1}{2} - \alpha\right){}^{t}\ddot{\mathbf{d}} + {}^{t+\Delta t}\ddot{\mathbf{d}}\alpha \right] \Delta t^2 \end{cases} \tag{34}$$

where $\Delta t$ is the time step for the temporal integration and $\alpha$ and $\delta$ are the unknown coefficients associated with the accuracy of the integration.

Additionally, the equilibrium equation at time $t + \Delta t$ following should also be utilized,

$$\mathbf{C}^{t+\Delta t}\dot{\mathbf{d}} + \mathbf{M}^{t+\Delta t}\ddot{\mathbf{d}} + \mathbf{K}^{t+\Delta t}\mathbf{d} = \mathbf{F} \tag{35}$$

These two parameters are utilized in this work since they will not add any numerical dampening effects to the result when $\beta = 1/2$ and $\alpha = 1/4$. Combining Equations (25) and (34), when $t + \Delta t$ time arrives, we can acquire,

$$\left(\frac{4}{\Delta t^2}\mathbf{M} + \frac{2}{\Delta t}\mathbf{C} + \mathbf{K}\right)^{t+\Delta t}\mathbf{d} = \mathbf{M}\left(\frac{4}{\Delta t^2}{}^t\mathbf{d} + \frac{4}{\Delta t}{}^t\dot{\mathbf{d}} + {}^t\ddot{\mathbf{d}}\right) + \mathbf{C}\left(\frac{2}{\Delta t}{}^t\mathbf{d} + {}^t\dot{\mathbf{d}}\right) \tag{36}$$

After that, the whole numerical solution can be obtained by repeatedly applying Equations (34) and (35).

## 3. Analysis of 2D Examples

First, we used 2D examples to evaluate and analyze the numerical convergence and accuracy of E-FEM.

### 3.1. The Cantilever Beam

Unless otherwise stated, the global standard unit system is the foundation for all physical units used in the current work. The numerical experiment of 2D free vibration analysis was performed with a cantilever beam model (see Figure 2). We studied various behaviors of E-FEM elements of a 2D cantilever beam with height D and length L. The inputs used in the calculation included: Young's modulus $E = 2.1 \times 10^4$ kgf/mm$^2$, thickness $t = 1$ mm, Poisson's ratio $v = 0.3$, D $= 10$ mm, L $= 100$ mm, mass density $\rho = 8.0 \times 10^{-10}$ kgfs$^2$/mm$^4$. This was obtained by dividing the cantilever beam of Figure 2 into three types of meshes with different precision in Figure 3. Liu and Nguyen-Thoi, and Nagashima have already examined this issue using NS-FEM and ES-FEM [32,33]; Liu and Nguyen-Thoi used the node-by-node meshless (NBNM) approach; Liu and Gu utilized the Gaussian radial function [34] and local radial point interpolation method (LRPIM) with multi-quadrics (MQ) radial function.

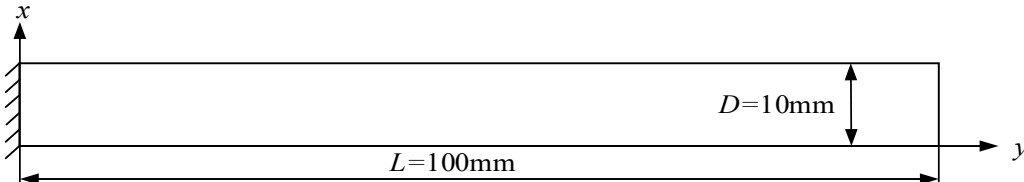

**Figure 2.** 2D cantilever beam.

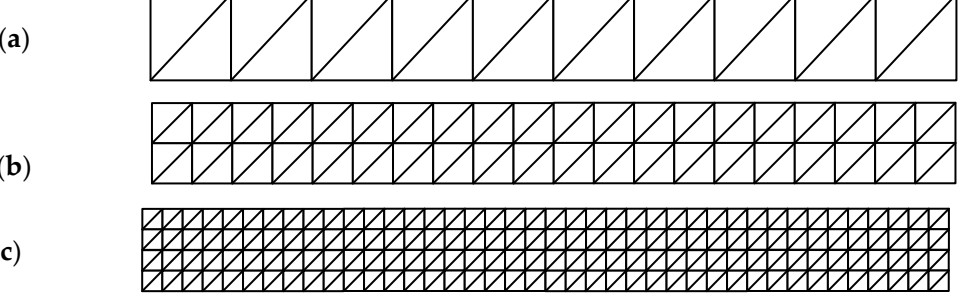

**Figure 3.** Triangle mesh for cantilever beam in Figure 4: (**a**) Mesh a (10 × 1); (**b**) Mesh b (20 × 2); (**c**) Mesh c (40 × 4).

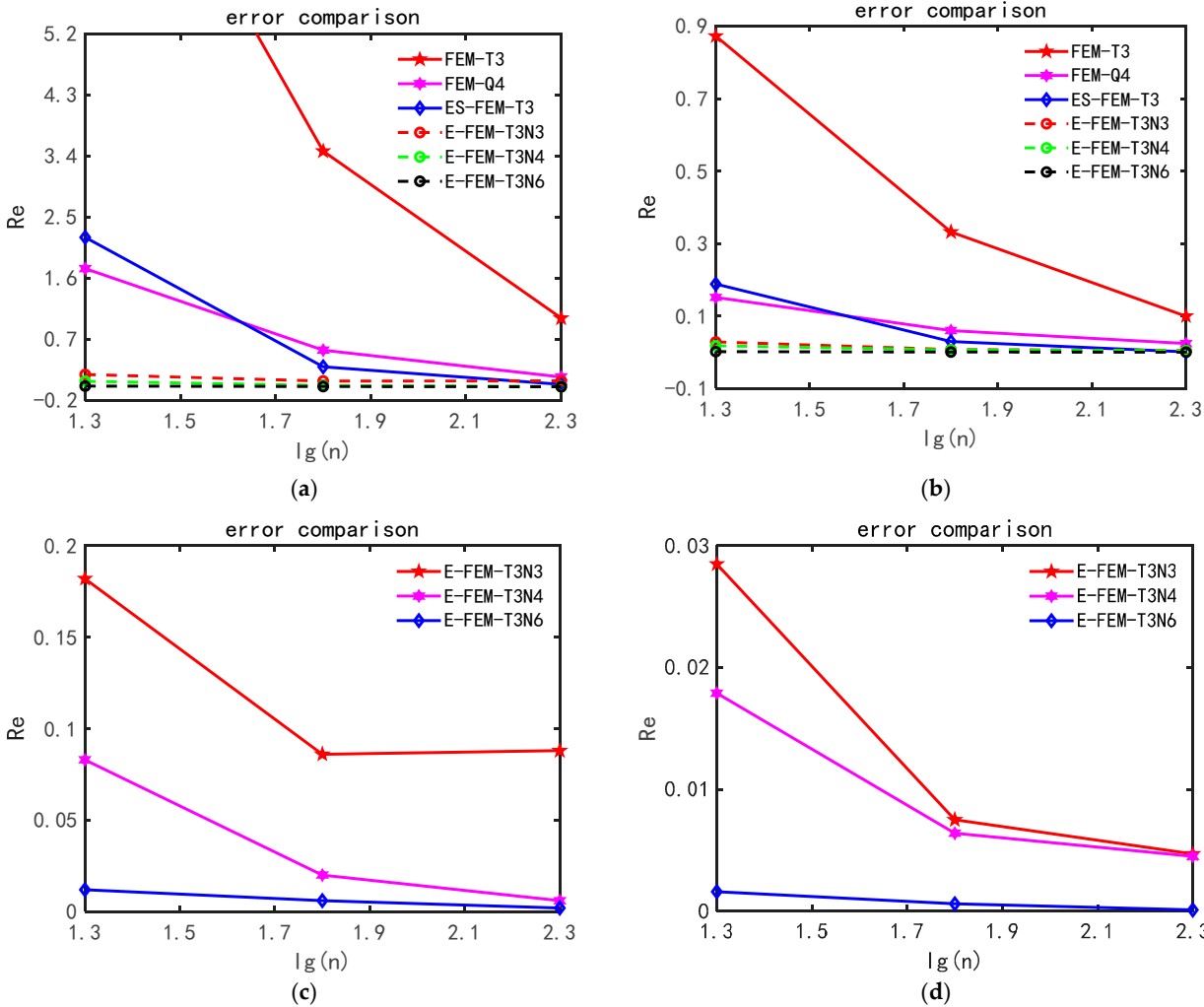

**Figure 4.** Comparison of calculation frequency accuracy of two modes: (**a**) Mode two; (**b**) Mode four; (**c**) Mode two; (**d**) Mode four.

### 3.1.1. Convergence Study

FEM-T3, E-FEM-T3, E-FEM-T3N4, and E-FEM-T3N6 were used in MATLAB to evaluate and calculate Mesh a, b, and c. Retrieving the frequencies of Tables 1–3 was possible in ten different modes. The following is the expression for the relative error of natural frequency, where n refers to the number of nodes, which is used to assess precision and convergence.

**Table 1.** Results of frequencies (Hz) of grid A ($10 \times 1$) (22 nodes, 20 elements).

| Order | FEM-T3 | FEM-Q4 | ES-FEM-T3 | E-FEM-T3N3 | E-FEM-T3N4 | E-FEM-T3N6 | References |
|-------|--------|--------|-----------|------------|------------|------------|------------|
| $f_1$ | 1708 | 992 | 1048 | 826 | 826 | 823 | 822 |
| $f_2$ | 9689 | 5791 | 6018 | 4997 | 4973 | 4938 | 4932 |
| $f_3$ | 12,908 | 12,834 | 12,833 | 12,834 | 12,833 | 12,827 | 12,824 |
| $f_4$ | 24,331 | 14,830 | 15,177 | 13,311 | 13,174 | 13,014 | 12,993 |
| $f_5$ | 39,193 | 26,183 | 26,362 | 24,523 | 24,111 | 23,670 | 23,611 |
| $f_6$ | 42,944 | 38,140 | 37,724 | 37,946 | 37,051 | 36,149 | 36,010 |
| $f_7$ | 64,559 | 38,824 | 38,559 | 38,482 | 38,473 | 38,453 | 38,444 |
| $f_8$ | 67,691 | 51,924 | 50,349 | 53,047 | 51,413 | 49,865 | 49,578 |
| $f_9$ | 90,810 | 62,345 | 60,827 | 64,059 | 64,032 | 63,990 | 63,913 |
| $f_{10}$ | 98,302 | 64,846 | 61,520 | 69,457 | 66,800 | 64,440 | 63,975 |

**Table 2.** Results of frequencies (Hz) grid B (20 × 2) (63 nodes, 80 elements).

| Order | FEM-T3 | FEM-Q4 | ES-FEM-T3 | E-FEM-T3N3 | E-FEM-T3N4 | E-FEM-T3N6 | References |
|-------|--------|--------|-----------|------------|------------|------------|------------|
| $f_1$ | 1120 | 870 | 853 | 824 | 824 | 823 | 822 |
| $f_2$ | 6644 | 5199 | 5078 | 4945 | 4942 | 4935 | 4932 |
| $f_3$ | 12,852 | 12,830 | 12,828 | 12,828 | 12,827 | 12,825 | 12,824 |
| $f_4$ | 17,307 | 13,640 | 13,246 | 13,038 | 13,024 | 13,001 | 12,993 |
| $f_5$ | 31,173 | 24,685 | 23,783 | 23,729 | 23,687 | 23,629 | 23,611 |
| $f_6$ | 38,686 | 37,477 | 35,784 | 36,259 | 36,165 | 36,041 | 36,010 |
| $f_7$ | 47,342 | 38,378 | 38,298 | 38,455 | 38,454 | 38,448 | 38,444 |
| $f_8$ | 64,769 | 51,322 | 48,533 | 50,037 | 49,858 | 49,628 | 49,578 |
| $f_9$ | 65,365 | 63,585 | 61,527 | 63,996 | 63,991 | 63,975 | 63,913 |
| $f_{10}$ | 84,519 | 65,731 | 63,182 | 64,678 | 64,373 | 63,992 | 63,975 |

**Table 3.** Results of frequencies (Hz) of grid C (40 × 4) (205 nodes, 320 elements).

| Order | FEM-T3 | FEM-Q4 | ES-FEM-T3 | E-FEM-T3N3 | E-FEM-T3N4 | E-FEM-T3N6 | Reference |
|-------|--------|--------|-----------|------------|------------|------------|-----------|
| $f_1$ | 907 | 835 | 827 | 823 | 823 | 823 | 822 |
| $f_2$ | 5431 | 5004 | 4950 | 4936 | 4935 | 4933 | 4932 |
| $f_3$ | 12,834 | 12,827 | 12,826 | 12,825 | 12,825 | 12,824 | 12,824 |
| $f_4$ | 14,286 | 13,174 | 13,006 | 13,002 | 13,000 | 12,994 | 12,993 |
| $f_5$ | 25,949 | 23,926 | 23,554 | 23,631 | 23,626 | 23,614 | 23,611 |
| $f_6$ | 38,511 | 36,462 | 35,778 | 36,046 | 36,035 | 36,014 | 36,010 |
| $f_7$ | 39,612 | 38,431 | 38,408 | 38,448 | 38,447 | 38,445 | 38,444 |
| $f_8$ | 54,647 | 50,150 | 49,029 | 49,638 | 49,619 | 49,584 | 49,578 |
| $f_9$ | 64,236 | 63,883 | 62,867 | 63,980 | 63,969 | 63,919 | 63,913 |
| $f_{10}$ | 70,685 | 64,561 | 63,774 | 64,007 | 63,985 | 63,976 | 63,975 |

$$R_e = \left| \frac{f_{num} - f_{ref}}{f_{ref}} \right| \times 100\% \tag{37}$$

where the superscript "ref" means reference solution and the superscript "num" means numerical solution.

Then, using $lg(n)$ as the abscissa and $R_e$ as the ordinate, taking the former two modes as examples, the relative error diagram of Figure 4 was obtained by comparing it with a reference value. Tables 1–3 present a list of the frequencies calculated for the three meshes.

Figure 4 displays the natural frequency error of the second and fourth modes. In addition, the convergence comparison concluding the FEM-T3, E-FEM-T3N3, E-FEM-T3N4, E-FEM-T3N6, ES-FEM-T3, and FEM-Q4 are displayed in Figure 5. Based on the simulation results, the second and fourth natural frequencies of E-FEM-T3, which include E-FEM-T3N3, E-FEM-T3N4, and E-FEM-T3N6, have a considerably smaller error and a significant accuracy than FEM-Q4, E-FEM, and ES-FEM. Compared to other methods for the same mesh without adding new nodes, the performance of E-FEM in terms of calculation accuracy is rolled over with an inaccuracy of only about a thousandth. Additionally, under the same node and grid density, the accuracy of the E-FEM algorithm increases with increasing order q or the number of terms N.

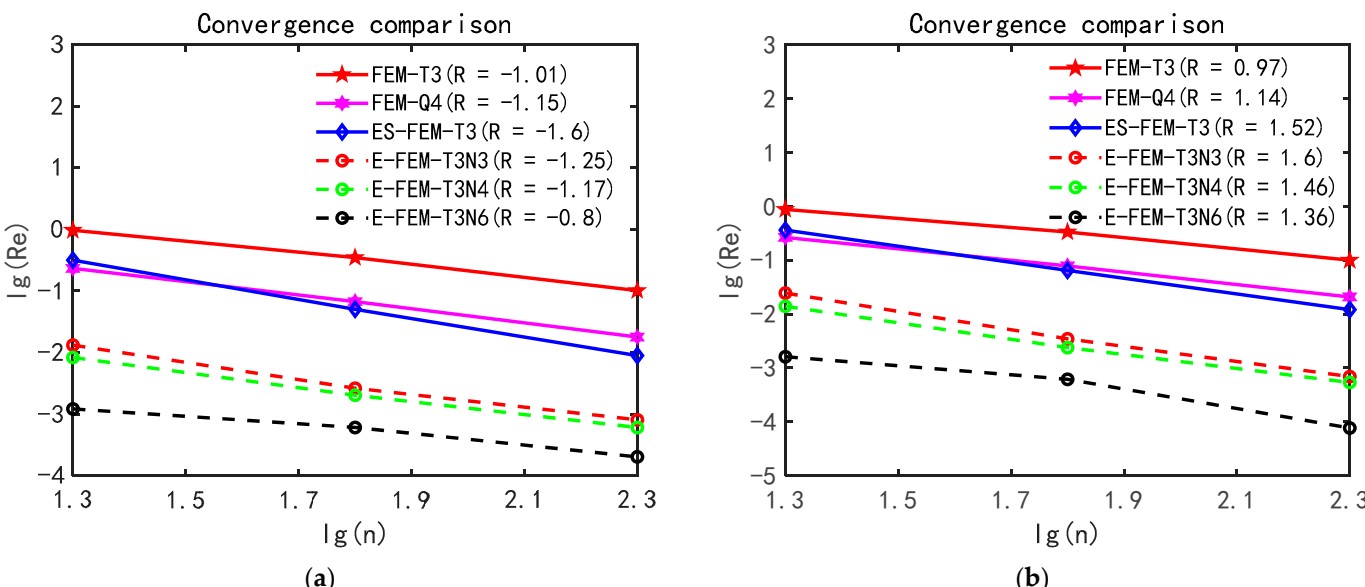

**Figure 5.** Convergence of the calculated frequency error of two modes: (**a**) Mode two; (**b**) Mode four.

Figure 6 displays the first to tenth eigenmodes concluded using the E-FEM element of Mesh. These model diagrams are superior compared to ES-FEM [13,17,35].

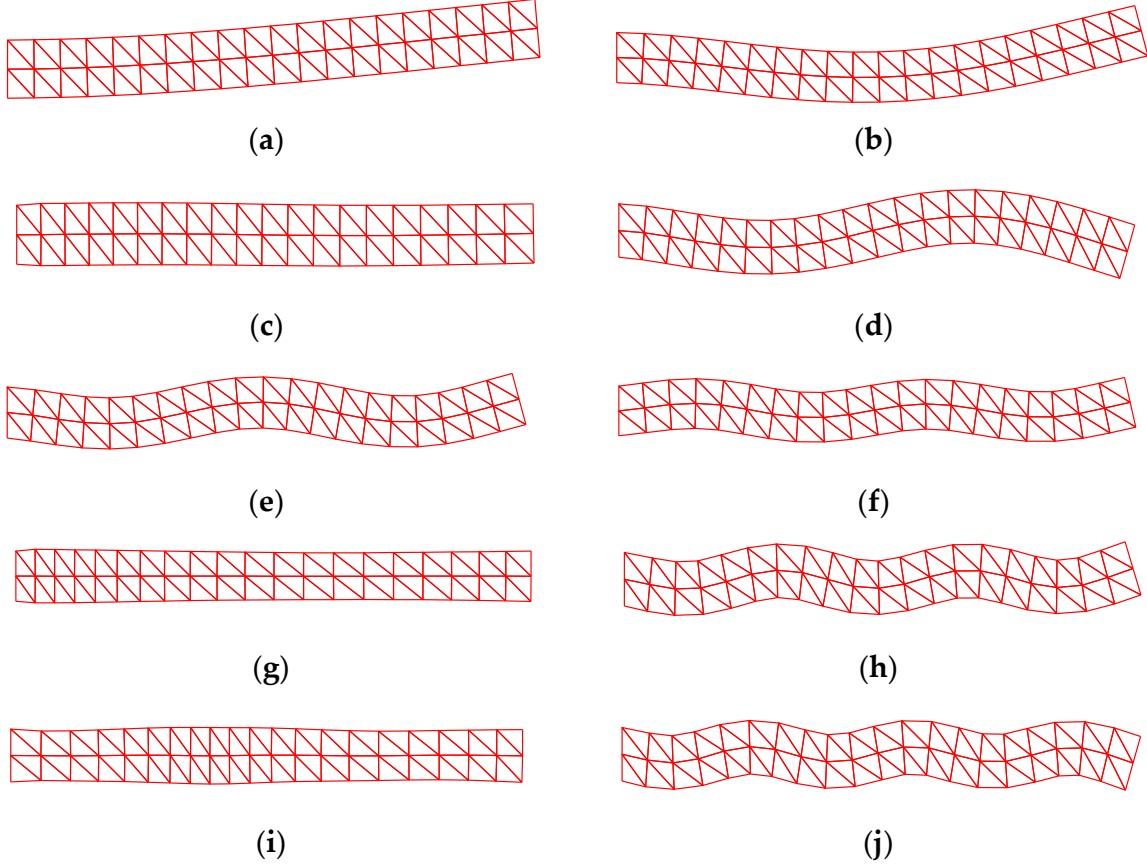

**Figure 6.** The top ten vibration modes of the cantilever beam calculated by E-FEM: (**a**) Mode one; (**b**) Mode two; (**c**) Mode three; (**d**) Mode four; (**e**) Mode five; (**f**) Mode six; (**g**) Mode seven; (**h**) Mode eight; (**i**) Mode nine; (**j**) Mode ten.

### 3.1.2. Grid Distortion Sensitivity Study

We give the distorted element mesh in Figure 7, and d is the distorted displacement.

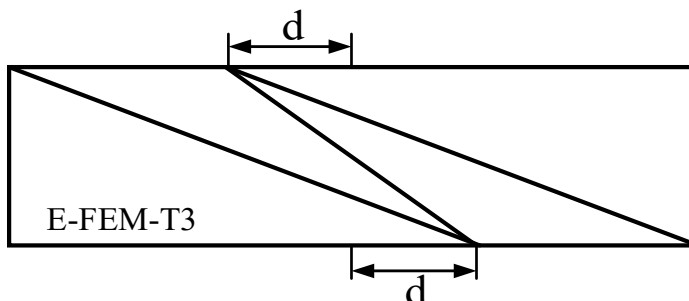

**Figure 7.** Distorted element mesh.

We can learn the following information by comparing relative errors of calculated frequencies, as shown in Figure 8 and Table 4. First, as deformation increases, the numerical error of the E-FEM-T3 does not vary significantly, whereas the numerical error of the FEM-T3 and FEM-Q4 varies. Secondly, the E-FEM-T3 curve is the lowest, with the smallest relative error. Thirdly, the error curves formed by FEM-T3 and FEM-Q4 follow a similar trend, with the error increasing when distortion parameters are increased. Fourthly, the curves of E-FEM-T3 are close to parallel lines, indicating that grid deformation has little effect on the error [36,37].

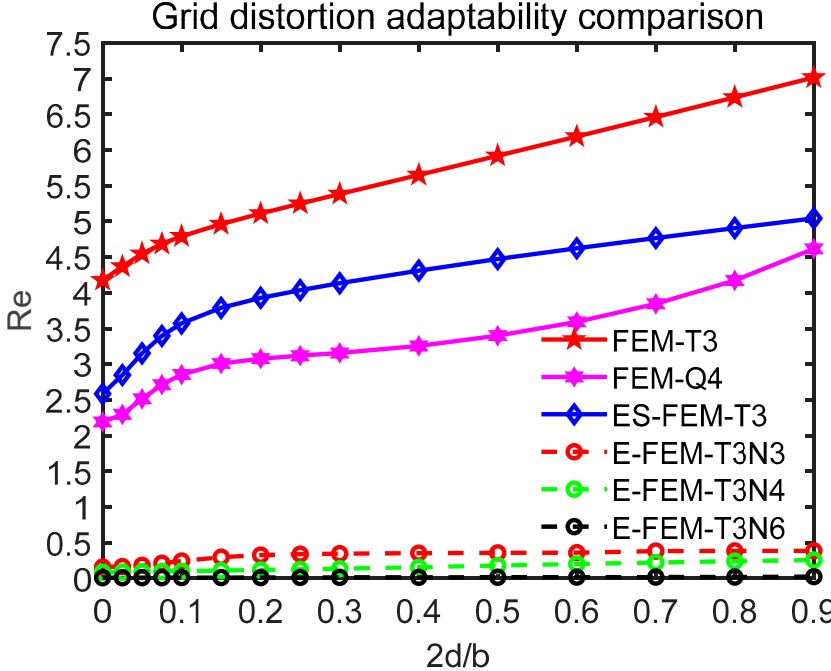

**Figure 8.** Calculated frequency error of the first mode for the distortion sensitivity test.

Figure 9 shows the two distorted grids established to assess the effect of grid quality on relative error. Three methods, FEM-T3, ES-FEM-T3, and E-FEM-T3, are shown in Tables 5 and 6. The enriched K-matrix and M-matrix of proposed E-FEM-T3 often yield better accuracy than that of FEM-T3 and Quad 4. Even for distorted grids, good results can be obtained using E-FEM-T3 elements. This is a significant benefit of the E-FEM-T3 method. The result is valuable for the practical application of E-FEM-T3. Unlike the E-FEM method, E-FEM-T3 can yield better results; however, additional nodes are necessary to increase the stiffness matrix.

**Table 4.** Grid distortion sensitivity calculation results.

| $2d/b$ | FEM-T3 | FEM-Q4 | ES-FEM-T3 | E-FEM-T3N3 | E-FEM-T3N4 | E-FEM-T3N6 | References |
|---|---|---|---|---|---|---|---|
| 0.000 | 4256 | 2623 | 2947 | 870 | 851 | 829.0 | 822 |
| 0.025 | 4413 | 2710 | 3163 | 875 | 853 | 829.5 | 822 |
| 0.050 | 4560 | 2889 | 3413 | 890 | 856 | 829.7 | 822 |
| 0.075 | 4674 | 3052 | 3614 | 913 | 860 | 830.0 | 822 |
| 0.100 | 4762 | 3169 | 3758 | 939 | 863 | 830.3 | 822 |
| 0.150 | 4902 | 3296 | 3937 | 983 | 870 | 830.8 | 822 |
| 0.200 | 5023 | 3352 | 4052 | 1007 | 877 | 831.5 | 822 |
| 0.250 | 5137 | 3386 | 4141 | 1018 | 884 | 832.3 | 822 |
| 0.300 | 5248 | 3417 | 4221 | 1024 | 892 | 833.0 | 822 |
| 0.400 | 5468 | 3498 | 4366 | 1030 | 909 | 834.0 | 822 |
| 0.500 | 5688 | 3617 | 4499 | 1033 | 928 | 835.0 | 822 |
| 0.600 | 5910 | 3776 | 4623 | 1035 | 946 | 835.7 | 822 |
| 0.700 | 6134 | 3984 | 4741 | 1037 | 963 | 836.4 | 822 |
| 0.800 | 6361 | 4252 | 4856 | 1039 | 978 | 837.2 | 822 |
| 0.900 | 6591 | 4617 | 4968 | 1040 | 992 | 841.2 | 822 |

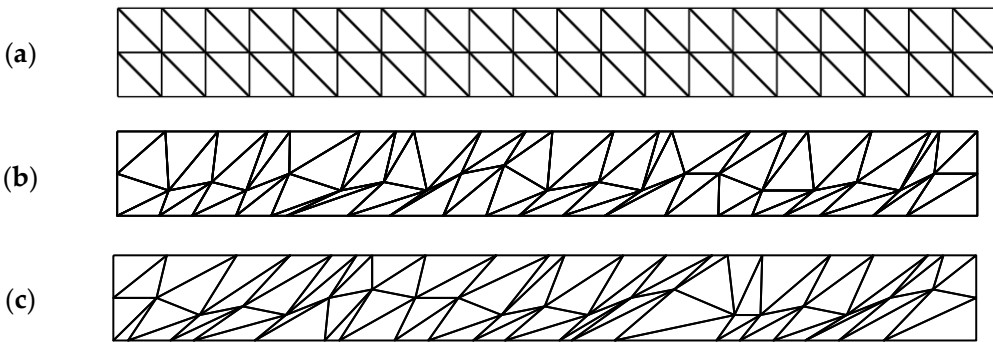

(**a**)

(**b**)

(**c**)

**Figure 9.** The distorted grids of Figure 4 (mesh: 20 × 2):(**a**) Mesh A; (**b**) Mesh B; (**c**) Mesh C.

**Table 5.** Computed frequencies (Hz) of distorted grids in Figure 9b.

| Order | FEM-T3 | ES-FEM-T3 | E-FEM-T3N3 | E-FEM-T3N4 | E-FEM-T3N6 | References |
|---|---|---|---|---|---|---|
| $f_1$ | 1376 | 947 | 825 | 824 | 823 | 822 |
| $f_2$ | 8554 | 5663 | 4960 | 4946 | 4936 | 4932 |
| $f_3$ | 12,870 | 12,827 | 12,828 | 12,828 | 12,825 | 12,824 |
| $f_4$ | 21,883 | 14,726 | 13,112 | 13,036 | 13,004 | 12,993 |
| $f_5$ | 38,777 | 27,242 | 23,975 | 23,715 | 23,637 | 23,611 |
| $f_6$ | 40,231 | 38,194 | 36,927 | 36,214 | 36,059 | 36,010 |
| $f_7$ | 62,931 | 39,798 | 38,458 | 38,456 | 38,449 | 38,444 |
| $f_8$ | 67,377 | 54,754 | 51,329 | 49,958 | 49,668 | 49,578 |
| $f_9$ | 86,597 | 62,901 | 64,010 | 63,996 | 63,982 | 63,913 |
| $f_{10}$ | 94,872 | 72,378 | 66,640 | 64,569 | 64,080 | 63,975 |

In practical engineering problems, grid distortion will also have a certain impact on calculation accuracy. The next sections further explore the capacity of the E-FEM-T3 to resist grid distortion.

Figure 10 reveals a similar number of nodes in both uniform and distorted meshes, in which a higher degree of irregularity causes a more distorted mesh. Tables 5 and 6 show the top 10 natural frequency outcomes from various elements using two distorted meshes. The tables also include matching computation results from the uniform mesh and reference solutions. The findings indicate that using the distorted mesh pattern will significantly reduce the precision of the standard linear elements (FEM-T3 and FEM-Q4).

Although the estimated numerical solutions are excellent when using the deformed mesh, the present E-FEM-T3 method performs the best among all the factors considered. With a larger irregularity indication, these conclusions become increasingly more obvious. The E-FEM performs significantly better than the conventional FEM and ES-FEM. This indicates that, unlike other elements, the current E-FEM-T3 has a larger tolerance for mesh distortion.

**Table 6.** Computed frequencies (Hz) of distorted grids in Figure 9c.

| Order | FEM-T3 | ES-FEM-T3 | E-FEM-T3N3 | E-FEM-T3N4 | E-FEM-T3N6 | References |
|---|---|---|---|---|---|---|
| $f_1$ | 1584 | 1029 | 825 | 824 | 822 | 822 |
| $f_2$ | 9522 | 6227 | 4978 | 4944 | 4936 | 4932 |
| $f_3$ | 12,887 | 12,826 | 12,828 | 12,828 | 12,825 | 12,824 |
| $f_4$ | 24,124 | 15,611 | 13,283 | 13,030 | 13,002 | 12,993 |
| $f_5$ | 39,120 | 29,557 | 24,373 | 23,704 | 23,635 | 23,611 |
| $f_6$ | 47,170 | 38,019 | 37,259 | 36,235 | 36,078 | 36,010 |
| $f_7$ | 67,566 | 46,072 | 38,460 | 38,455 | 38,448 | 38,444 |
| $f_8$ | 74,501 | 61,371 | 53,065 | 49,995 | 49,708 | 49,578 |
| $f_9$ | 97,634 | 63,573 | 64,021 | 63,993 | 63,981 | 63,913 |
| $f_{10}$ | 102,709 | 76,854 | 70,025 | 64,674 | 64,180 | 63,975 |

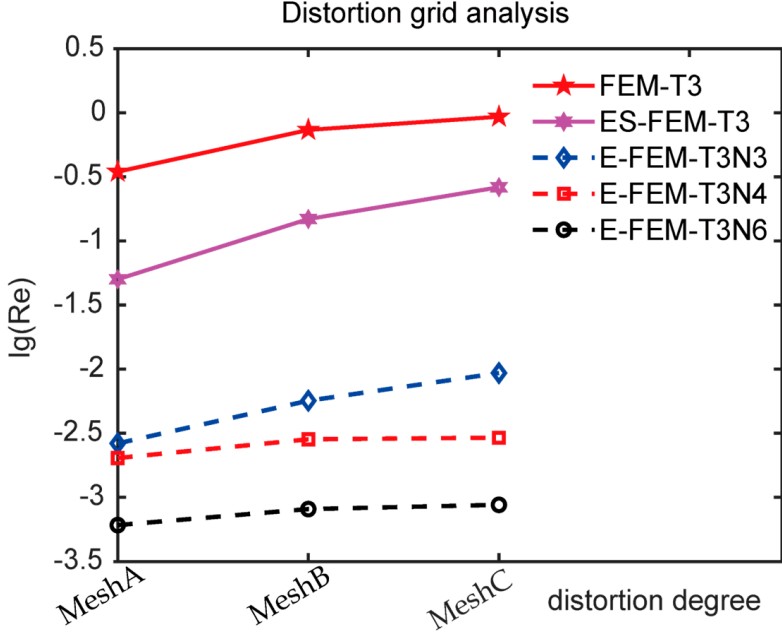

**Figure 10.** Grid distortion adaptability comparison.

Figure 11 shows the related E-FEM and FEM element solution times, and the relative imprecision of the two natural frequencies. At the second natural frequency, the E-FEM-T3N3 element converges at a rate of 0.82 slower than the FEM. Regarding the convergence rate at the fourth natural frequency, the FEM element performs better than the E-FEM element, standing at 1.71. As shown, the convergence rate of error relative to computation time for the three E-FEM algorithms reduces as N increases. The accuracy of distorted and standard meshes can be improved using a similar mesh to the E-FEM-T3 element. The E-FEM-T3 increases the computation time by increasing the bandwidth of the total stiffness matrix. Therefore, there is a significant urgency to strike a good balance between accuracy and computational speed for the E-FEM-T3 approach.

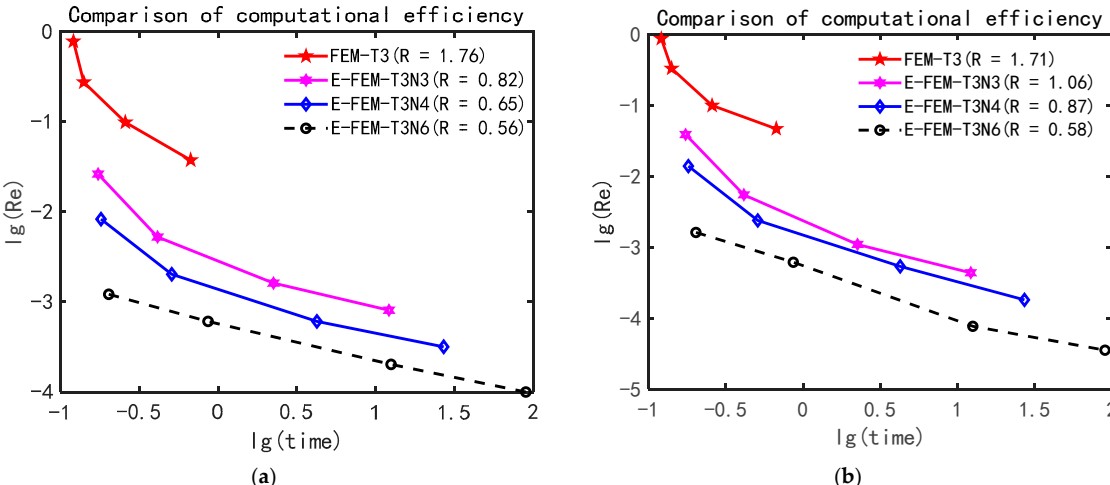

**Figure 11.** The comparison of computational efficiency for two modes: (**a**) Mode two; (**b**) Mode four.

### 3.1.3. Forced Vibration Study

The E-FEM was used to investigate a benchmark cantilever beam problem using the Newmark method for time stepping. A tip harmonic loading in the *y*-direction, $f(t) = \cos \omega_f t$, was applied to the beam. The numerical parameters used to consider the plane strain problem included: $E = 0.1$, $H = 1.0$, $L = 4.0$, $v = 0.3$, $\alpha = 0.005$, $\beta = 0.272$, $t = 1.0$. The model comprised 100 units and 126 nodes, computing with an amplitude time step of $\Delta t = 1$. By analyzing the performance of E-FEM-T3, FEM-Q4, ES-FEM-T3, and FEM-T3 in this model, the vibration periods of E-FEM-T3 and FEM-Q4, ES-FEM-T3 and FEM-T3 were highly consistent (Figure 12a). Furtherly the amplitude of E-FEM-T3 was larger than that of ES-FEM-T3, that of ES-FEM-T3 was larger than that of FEM-Q4, and that of FEM-Q4 was larger than that of FEM-T3. The peak of each curve reached FEM-T3 (526.4), FEM-Q4 (809.6), ES-FEM-T3 (843.8), E-FEM-T3N3 (858.8), E-FEM-T3N4 (873.7) and E-FEM-T3N6 (883.7). The larger the amplitude, the softer the model. Therefore, the model accuracy of FEM-Q4 is higher than that of FEM-T3, with better convergence efficiency. Similarly, E-FEM-T3 achieves higher model accuracy and better convergence efficiency [38,39].

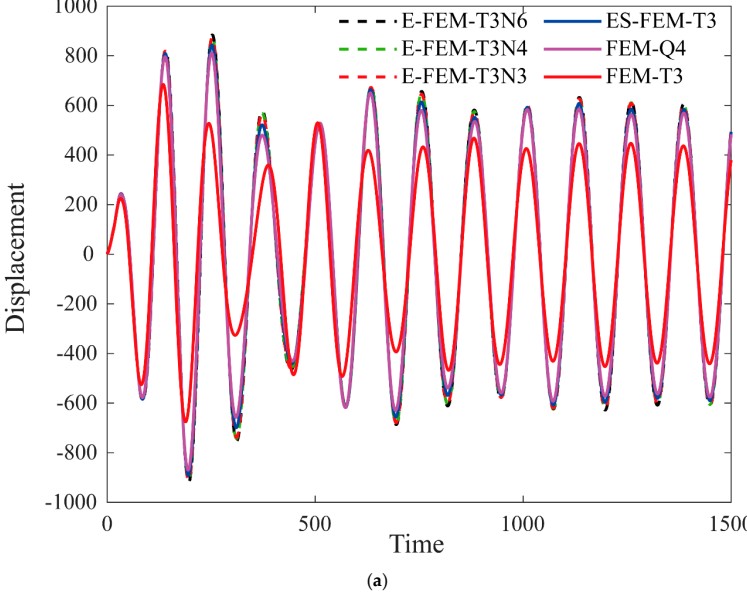

**Figure 12.** *Cont.*

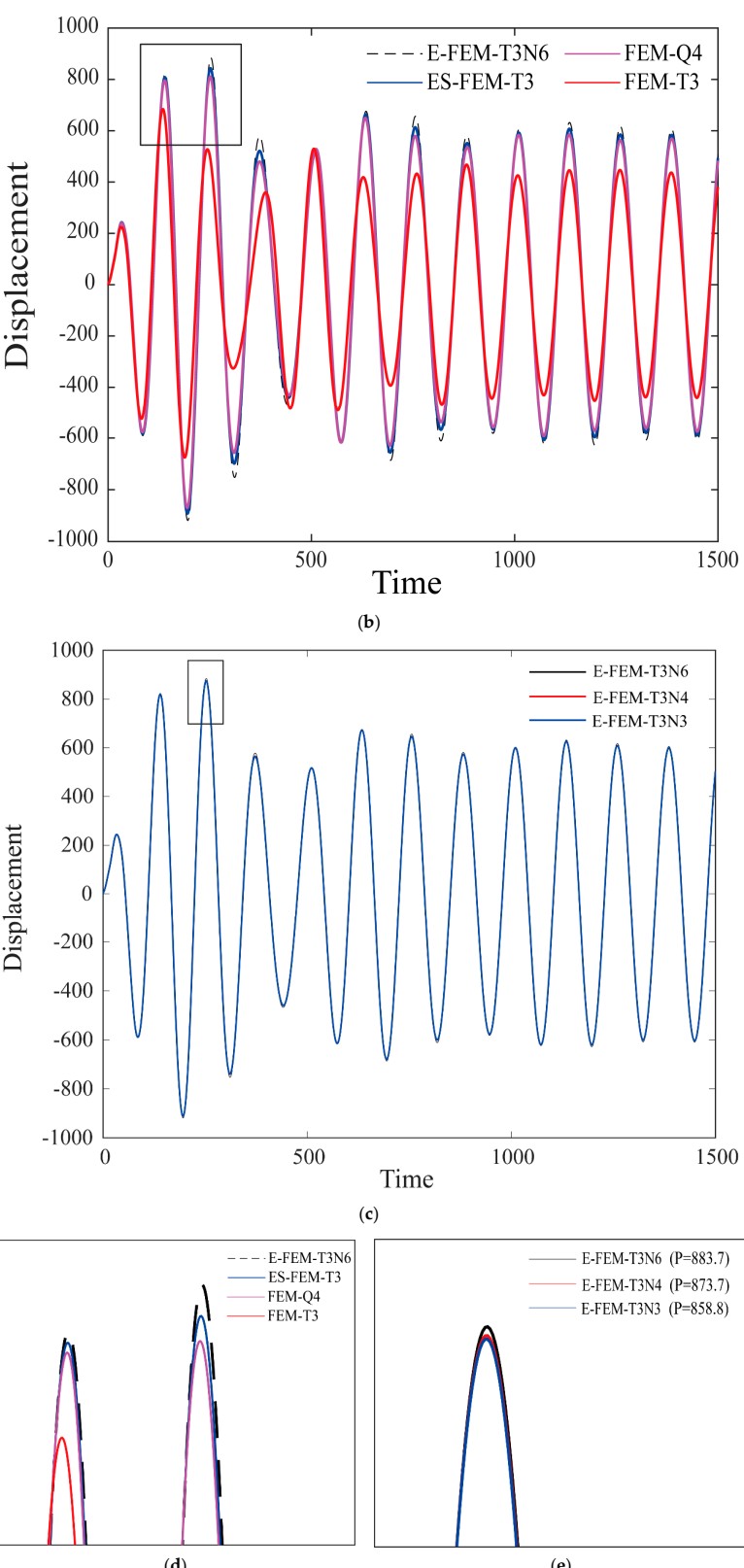

**Figure 12.** Comparison of transient responses of a cantilever beam by E-FEM-T3: (**a**) Results of all 6 algorithms; (**b**) Analysis of the performance of 4 algorithms; (**c**) Multi-basis function curve of forced vibration; (**d**) Partial enlarged view of Figure 12b; (**e**) Partial enlarged view of Figure 12c.

### 3.2. A Shear Wall

Figure 13 illustrates how Brabbia et al. solved a straightforward model of a four-opening shear wall with an assumed plane stress condition using the boundary element approach. In Figure 14, 559 uniformed nodes remedied the issue. Young's modulus $E = 1 \times 10^3$, Poisson's ratio $v = 0.2$, thickness $t = 1$, mass density $\rho = 1$, and these were the pertinent values. Figure 14 presents the grid computed with 952 cells and 559 nodes. As shown in Table 7, the E-FEM was used to calculate the natural frequencies of the first 10 modes. The E-FEM-T3 results are typically the ones closest to the reference solution since the generated natural frequencies are significantly bigger than the FEM-T3, which is excessively stiff. Considering that the natural frequencies are a reliable sign for determining the stiffness of a model, the results mentioned above verify that the E-FEM-T3 has extremely close-to-exact stiffness.

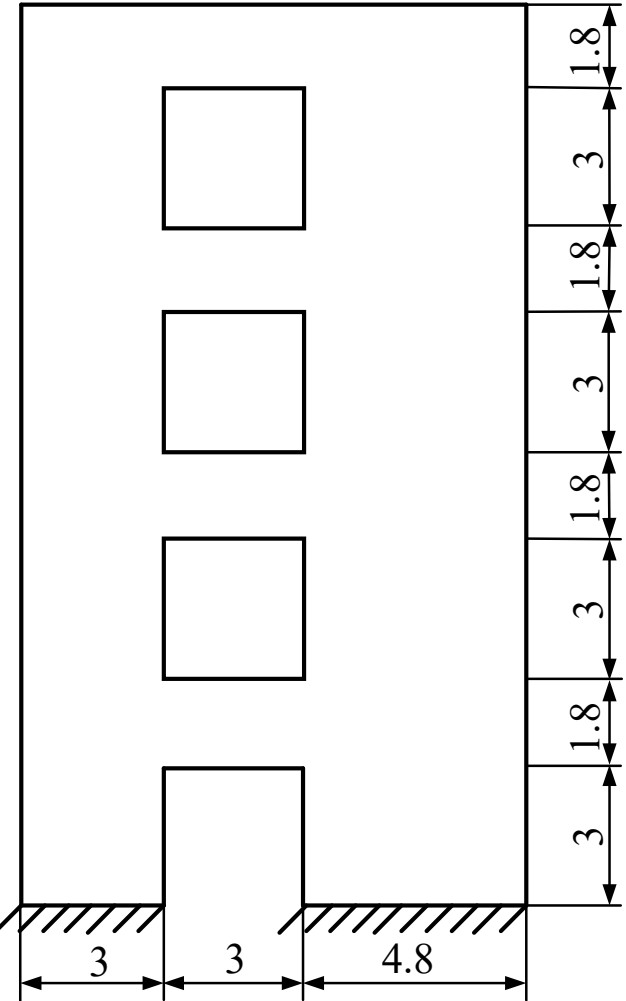

**Figure 13.** A shear wall.

**Table 7.** Natural frequencies (Hz) of a shear wall.

| Order | FEM-T3 | FEM-Q4 | ES-FEM-T3 | E-FEM-T3N3 | E-FEM-T3N4 | E-FEM-T3N6 | References |
|---|---|---|---|---|---|---|---|
| $f_1$ | 0.1081 | 0.1044 | 0.1032 | 0.1021 | 0.1019 | 0.1014 | 0.1011 |
| $f_2$ | 0.3681 | 0.3580 | 0.3553 | 0.3520 | 0.3515 | 0.3504 | 0.3497 |
| $f_3$ | 0.3855 | 0.3839 | 0.3836 | 0.3830 | 0.3828 | 0.3826 | 0.3825 |
| $f_4$ | 0.6312 | 0.6029 | 0.5916 | 0.5839 | 0.5823 | 0.5788 | 0.5767 |
| $f_5$ | 0.8094 | 0.7773 | 0.7677 | 0.7587 | 0.7579 | 0.7549 | 0.7532 |
| $f_6$ | 0.9503 | 0.9275 | 0.9214 | 0.9135 | 0.9119 | 0.9103 | 0.9094 |
| $f_7$ | 1.0352 | 1.0061 | 0.9983 | 0.9898 | 0.9882 | 0.9865 | 0.9857 |
| $f_8$ | 1.1459 | 1.1247 | 1.1158 | 1.1106 | 1.1045 | 1.1021 | 1.1007 |
| $f_9$ | 1.2045 | 1.1673 | 1.1552 | 1.1450 | 1.1434 | 1.1404 | 1.1389 |
| $f_{10}$ | 1.2276 | 1.1944 | 1.1844 | 1.1760 | 1.1750 | 1.1720 | 1.1724 |

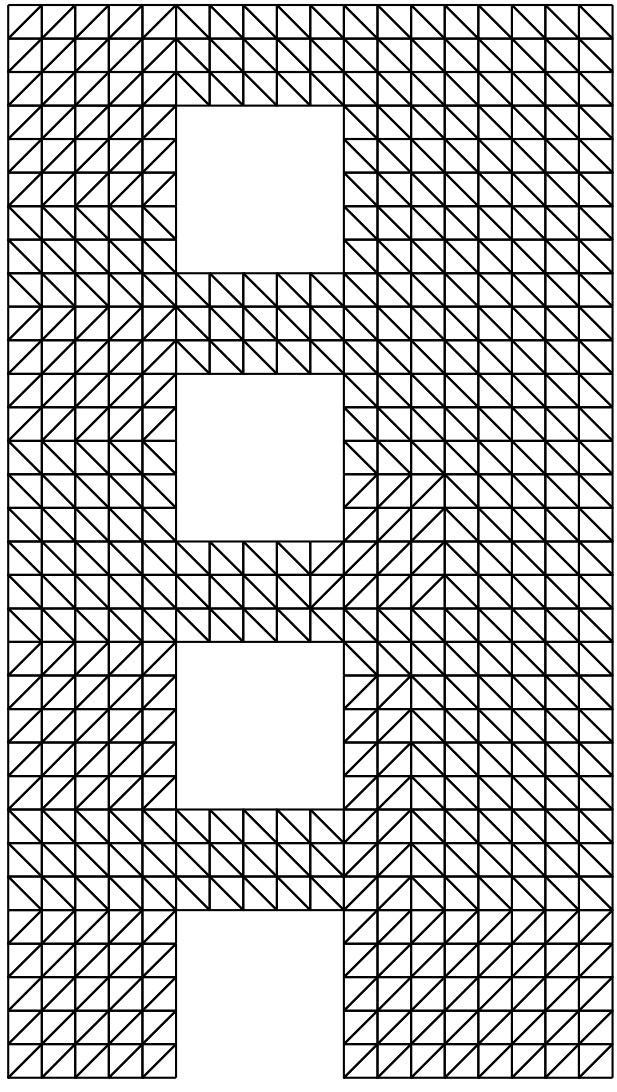

**Figure 14.** The grid of a shear wall grid.

We give the first 10 vibration modes of the shared wall computed using E-FEM in Figure 15. It is obvious that the proposed E-FEM indeed behaves very well in predicting the mode shape in free vibration analysis.

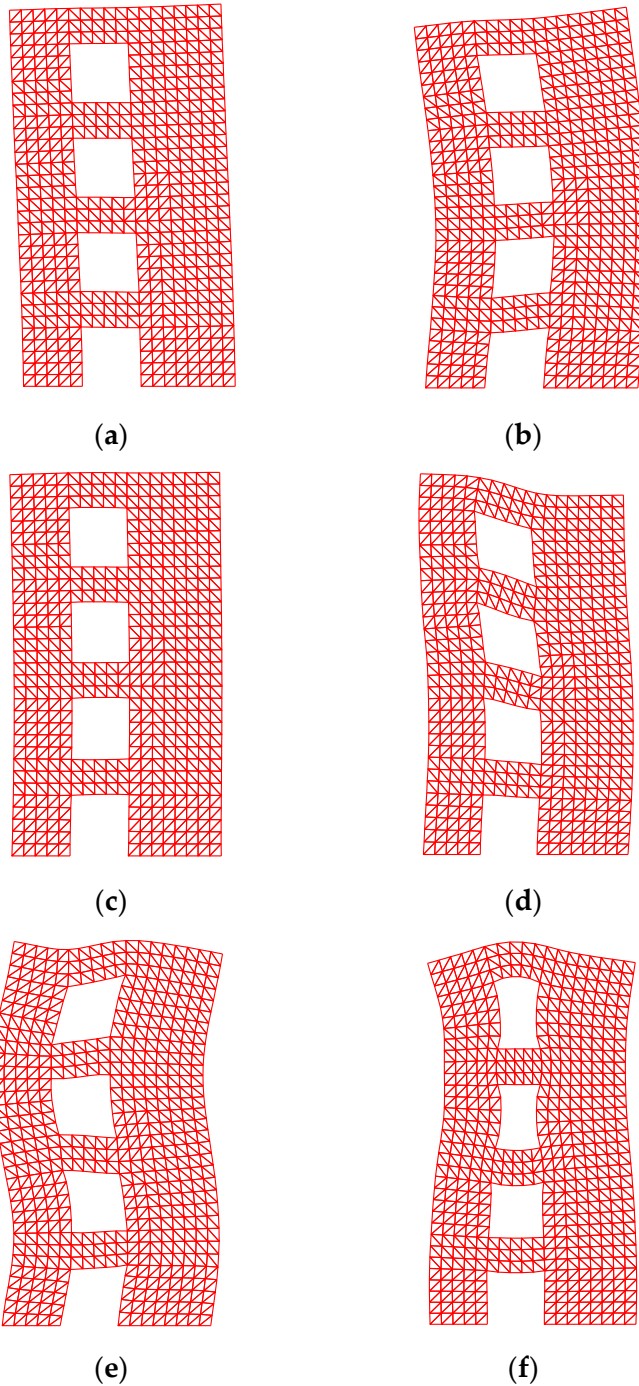

**Figure 15.** *Cont.*

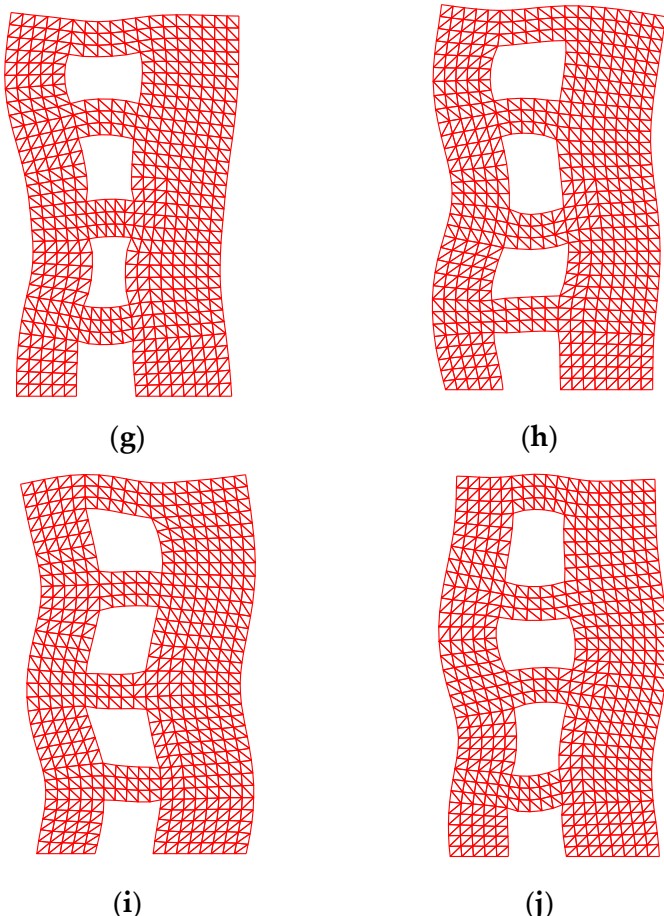

**Figure 15.** First 10 vibration modes of the shared wall computed using E-FEM: (**a**) Mode one; (**b**) Mode two; (**c**) Mode three; (**d**) Mode four; (**e**) Mode five; (**f**) Mode six; (**g**) Mode seven; (**h**) Mode eight; (**i**) Mode nine; (**j**) Mode ten.

### 3.3. A Connecting Rod

We established a simple connecting rod with a large and small opening, as shown in Figure 16. The free vibration of the rod was analyzed under the assumption of plane stress state; the relevant calculation parameters included mass density $\rho = 7800 \, \text{kg/m}^3$, Young's modulus E = 10 GPa, and Poisson's ratio $v = 0.3$. Two set directions existed on the left inner circle of the rod. As shown in Table 8, E-FEM-T3 produced results equivalent to those of the reference. Figure 17 shows the first 10-order eigenmodes obtained using the E-FEM-T3 element under Mesh A. These mode shape graphs efficiently match the ES-FEM-T3 plot.

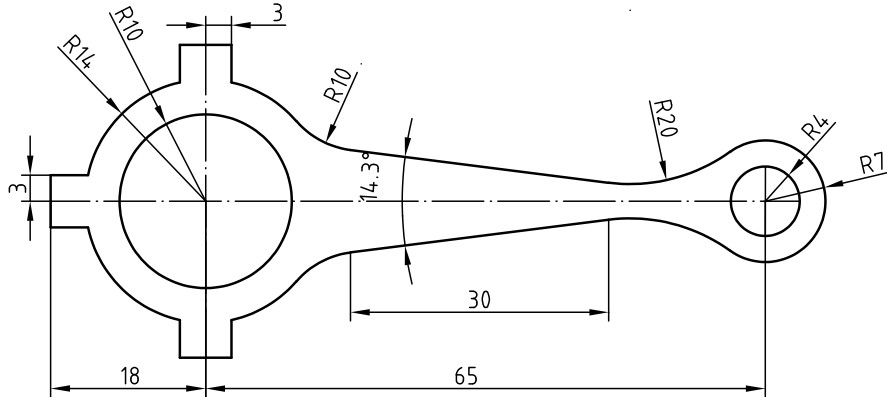

**Figure 16.** A geometric model and boundary conditions of an automobile connecting rod.

**Table 8.** First 10 natural frequencies (HZ) of a connecting rod.

| Order | FEM-T3 | FEM-Q4 | ES-FEM-T3 | E-FEM-T3N3 | E-FEM-T3N4 | E-FEM-T3N6 | References |
|---|---|---|---|---|---|---|---|
| $f_1$ | 158.4 | 144.6 | 140.9 | 144.8 | 142.1 | 141.2 | 140.7 |
| $f_2$ | 709.6 | 650.5 | 630.6 | 643.4 | 635.4 | 631.8 | 622.6 |
| $f_3$ | 1541.1 | 1535.2 | 1525.7 | 1555.4 | 1543.8 | 1535.4 | 1522.5 |
| $f_4$ | 1760.1 | 1644.7 | 1585.7 | 1578.5 | 1569.6 | 1565.5 | 1563.9 |
| $f_5$ | 3220.1 | 3028.4 | 2871.2 | 2897.3 | 2877.6 | 2857.3 | 2839.1 |
| $f_6$ | 3873.4 | 3797.9 | 3628.6 | 3586.1 | 3496.4 | 3484.1 | 3468.1 |
| $f_7$ | 4832.3 | 4503.2 | 4104.1 | 4259.5 | 4151.6 | 4040.5 | 3986.3 |
| $f_8$ | 5478.4 | 5274.7 | 4932.2 | 4996.4 | 4866.8 | 4846.3 | 4821.2 |
| $f_9$ | 5760.9 | 5473.5 | 4994.5 | 5017.9 | 4977.8 | 4957.9 | 4936.5 |
| $f_{10}$ | 6495.1 | 6177.8 | 5953.1 | 6251.6 | 6131.9 | 6081.5 | 6050.5 |

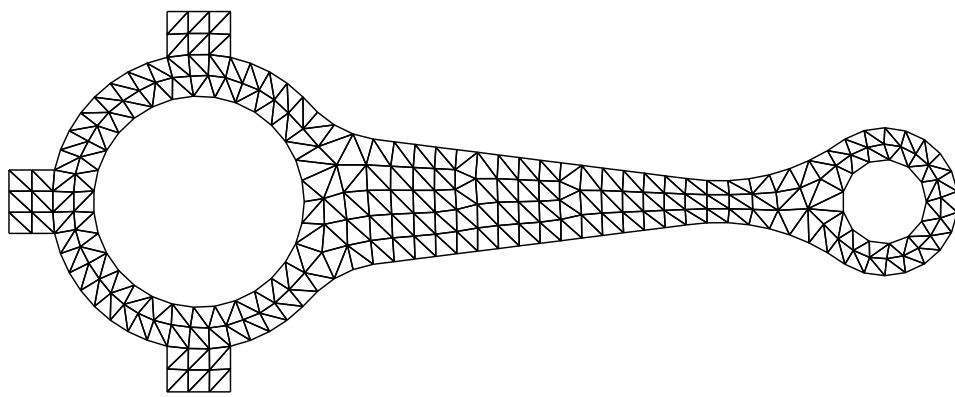

**Figure 17.** Mesh A of the connecting rod.

Figure 18 depicts the first 10 vibration modes of the shared wall computed using E-FEM. It is clear that the suggested E-FEM performs admirably in predicting mode shape in free vibration analysis.

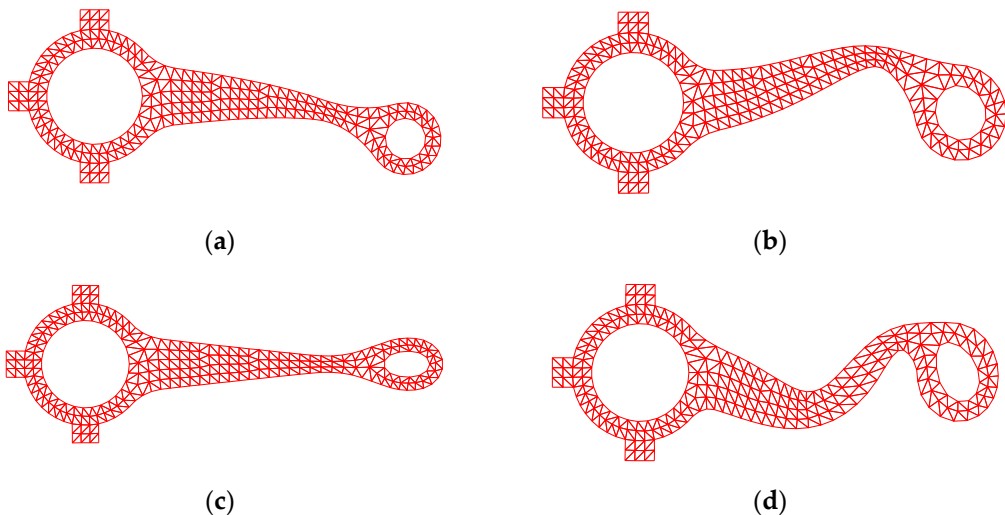

(**a**)      (**b**)

(**c**)      (**d**)

**Figure 18.** *Cont.*

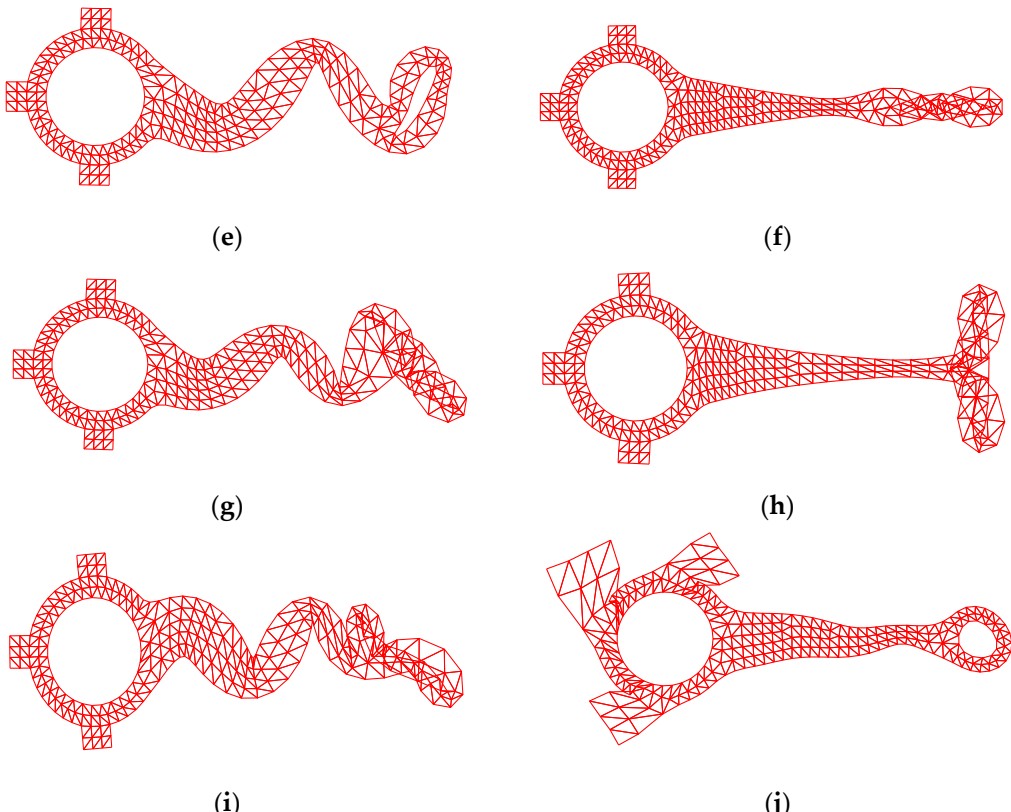

**Figure 18.** First 10 vibration modes of the connecting rod computed by E-FEM: (**a**) Mode one; (**b**) Mode two; (**c**) Mode three; (**d**) Mode four; (**e**) Mode five; (**f**) Mode six; (**g**) Mode seven; (**h**) Mode eight; (**i**) Mode nine; (**j**) Mode ten.

## 4. Analysis of 3D Examples

This section comprehensively explores the computational accuracy and computational efficiency of E-FEM in 3D models. We used two numerical examples: a cantilever beam structure and an engine connecting rod. The modes of the two numerical examples were calculated using the FEM, ES-FEM, and E-FEM, respectively.

### 4.1. The Cantilever Beam

Figure 19 shows the structure diagram of the cantilever beam with the geometric parameter of 0.12 m × 0.12 m × 0.72 m. The blue area represents the constrained surface. In the current analysis, we used the following material properties in calculations: elastic modulus E = 70 GPa, Poisson's ratio $v$ = 0.33, density $\rho$ = 2700 kg/m$^3$. First, we analyzed the calculation accuracy, which was discretized into a grid model with 752 nodes (Figure 19b). For comparison, the model was discretized into an extremely fine mesh (71,771 nodes and 25,200 elements) in ANSYS to obtain the reference solution.

Table 9 shows the first 15 modal eigenvalues of the cantilever beam, and the relative error is shown in Figure 20, indicating the smallest relative error of E-FEM, hence the highest calculation accuracy among the three methods.

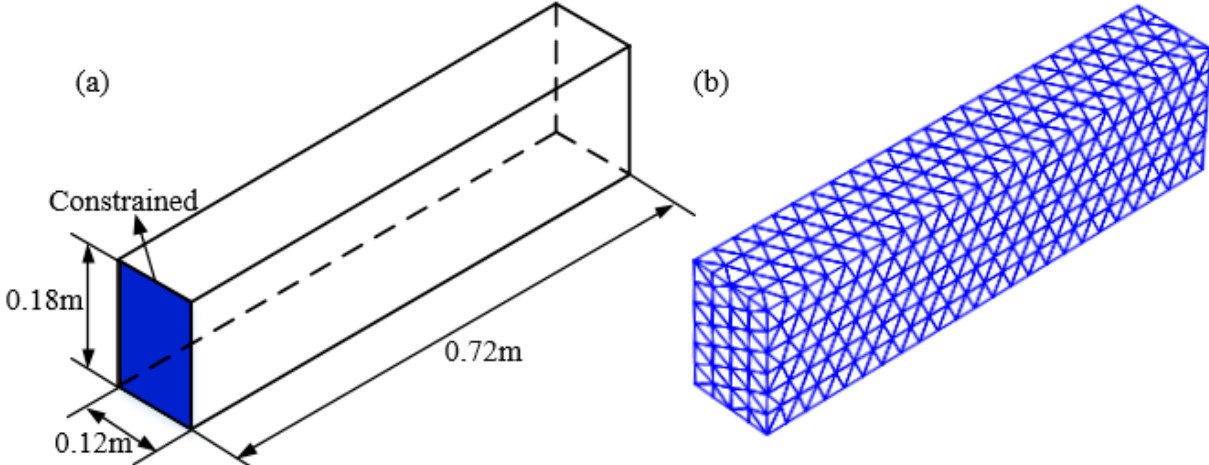

**Figure 19.** A cantilever model: (**a**) geometric model (**b**) mesh.

**Table 9.** Eigenvalues of the first 15 modes of the cantilever beam.

| Order | FEM | ES-FEM | E-FEM | Reference |
|---|---|---|---|---|
| $f_1$ | 207.73 | 190.47 | 189.30 | 188.67 |
| $f_2$ | 286.97 | 276.23 | 275.90 | 275.14 |
| $f_3$ | 1060.39 | 957.89 | 936.74 | 935.04 |
| $f_4$ | 1155.80 | 1069.11 | 1058.69 | 1055.70 |
| $f_5$ | 1445.77 | 1397.85 | 1393.07 | 1390.50 |
| $f_6$ | 1786.54 | 1782.74 | 1781.27 | 1779.90 |
| $f_7$ | 2823.05 | 2627.97 | 2597.55 | 2591.50 |
| $f_8$ | 3187.11 | 2876.26 | 2811.92 | 2806.40 |
| $f_9$ | 3318.56 | 3207.69 | 3189.32 | 3185.40 |
| $f_{10}$ | 4798.67 | 4488.68 | 4420.82 | 4412.30 |
| $f_{11}$ | 5323.91 | 4811.58 | 4692.26 | 46,820 |
| $f_{12}$ | 5343.23 | 5180.15 | 5137.22 | 5133.40 |
| $f_{13}$ | 5373.71 | 5307.97 | 5299.98 | 5296.20 |
| $f_{14}$ | 6963.50 | 6517.56 | 6397.92 | 6386.80 |
| $f_{15}$ | 7500.03 | 6764.03 | 6581.80 | 6564.10 |

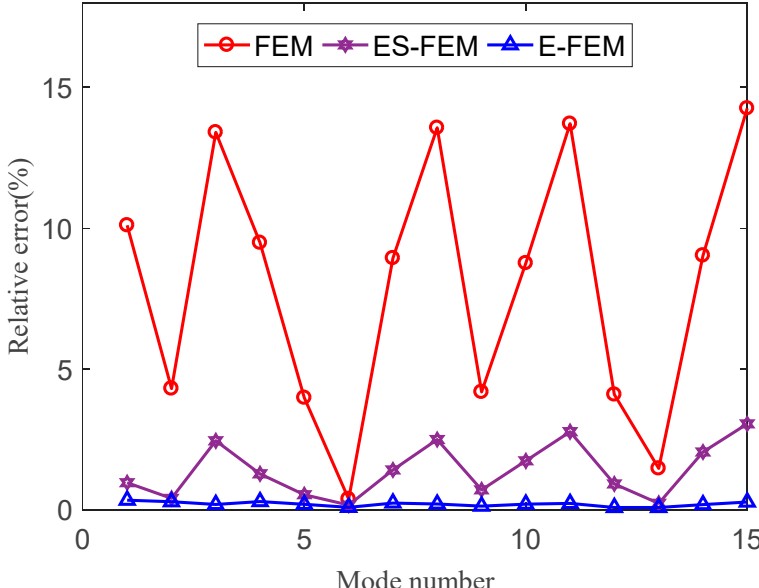

**Figure 20.** Relative error of first 15 modal eigenvalues of the cantilever beam.

We show the E-FEM calculation of the first 8 modes of a cantilever beam in Figure 21. It is apparent that the proposed E-FEM indeed behaves extremely well in predicting the mode shape.

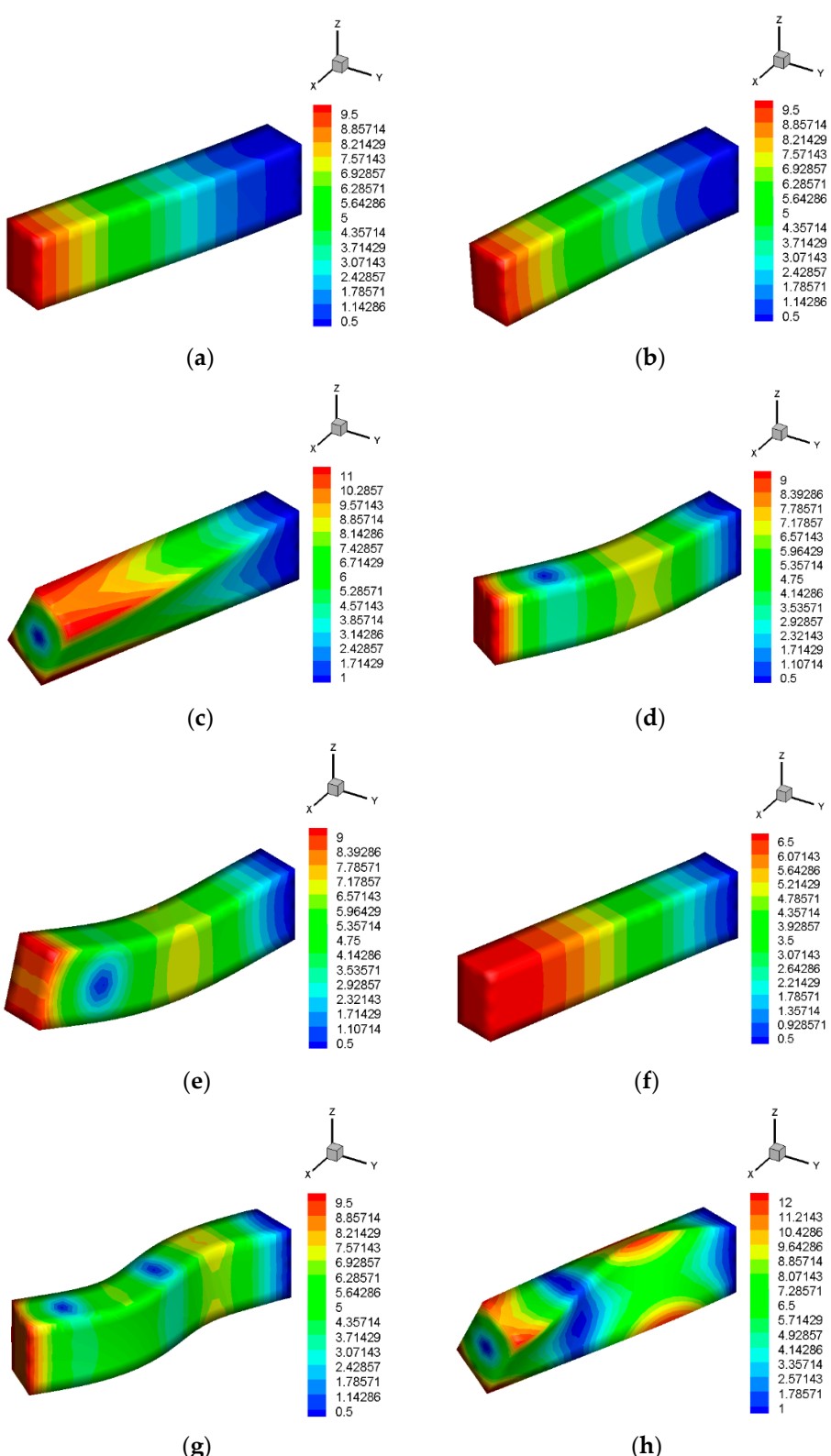

**Figure 21.** Calculation of the first 8 modes of a cantilever beam using E-FEM: (**a**) Mode one; (**b**) Mode two; (**c**) Mode three; (**d**) Mode four; (**e**) Mode five; (**f**) Mode six; (**g**) Mode seven; (**h**) Mode eight.

The cantilever beam was divided into four different sizes of grids to confirm the mesh robustness of E-FEM, with the number of nodes being 153, 399, 752, and 1135. The proposed E-FEM was used to analyze the natural frequency of the grid model, and the findings of FEM and ES-FEM are useful in the comparative analysis [31]. The fifth and 10-order natural frequencies were selected as research objects, and the relative error curve with the number of grid nodes was drawn (Figure 22). Therefore, based on Figure 22: (1) The three numerical methods are convergent; (2) Under a similar number of grid nodes, the calculation error of E-FEM is smaller than that of FEM and ES-FEM. The numerical example shows that E-FEM has good numerical convergence and accuracy.

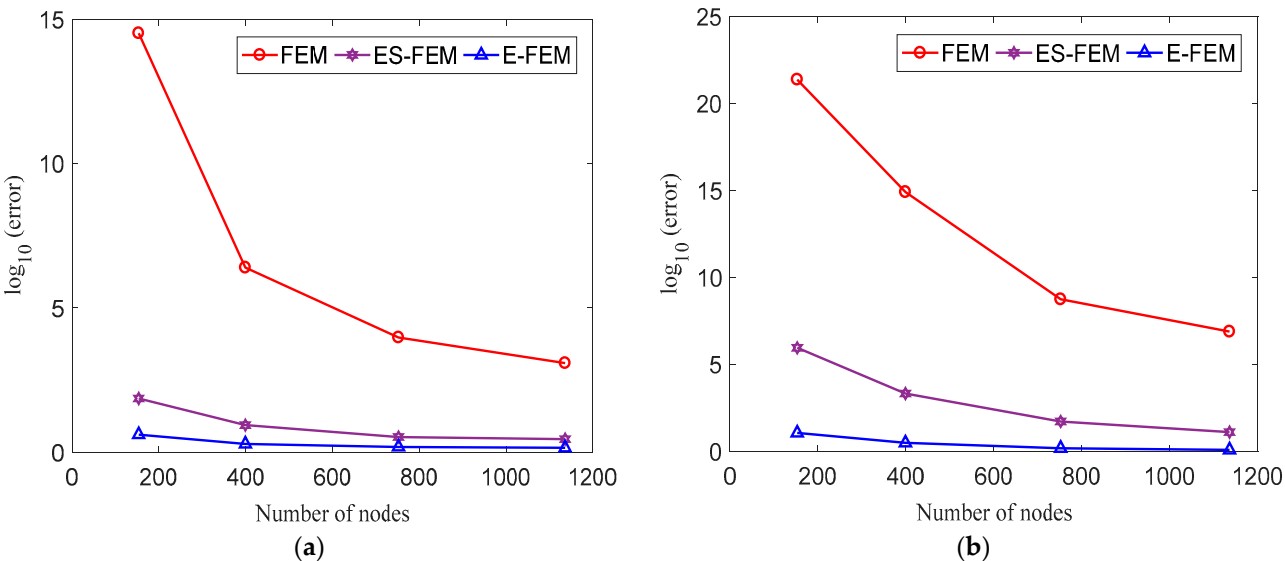

**Figure 22.** Convergence comparison: (**a**) Mode five; (**b**) Mode ten.

To investigate a numerical method, we need to analyze its computational efficiency. The same four kinds of grids are used. Figure 23 [32,33] shows the comparison of the calculation efficiency of three different methods (the ratio of the average error of the first 15 modes to time). It can be seen from the figure that the computational efficiency of E-FEM is higher than that of ES-FEM and FEM.

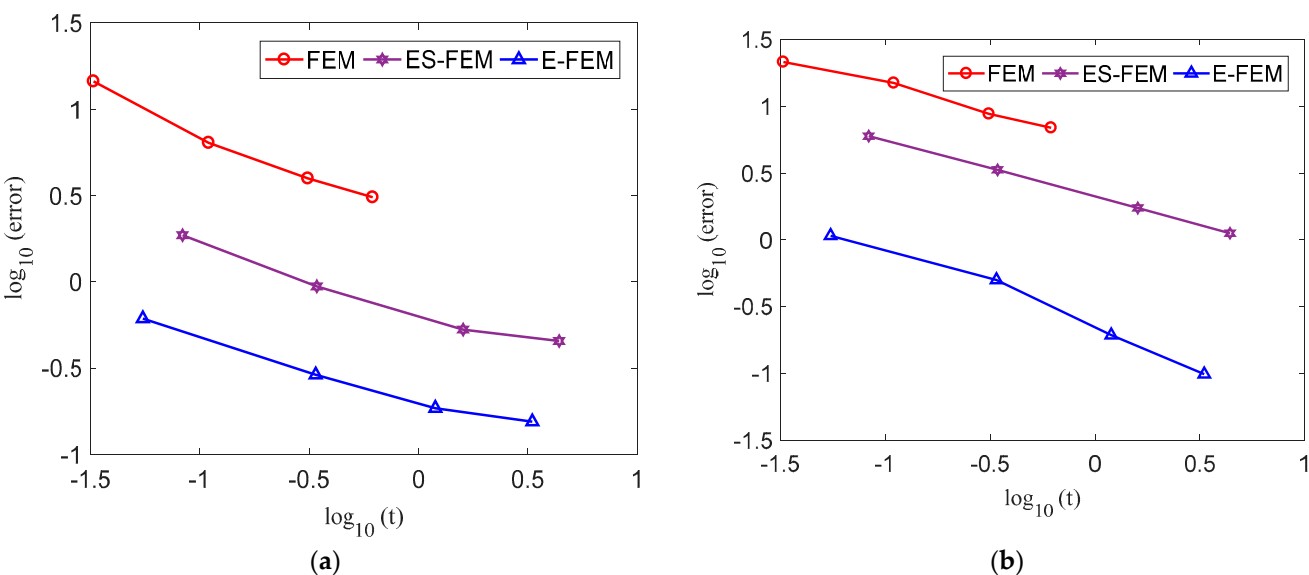

**Figure 23.** Comparison of computational efficiency: (**a**) Mode five; (**b**) Mode ten.

### 4.2. Engine Connecting Rod

In this section, we performed a modal analysis of an automobile engine connecting rod to confirm the applicability of the proposed method. Figure 24 displays the geometric model of the connecting rod, and the small end of the connecting rod is a fixed constraint. The material parameters include E = 210 GPa, $v = 0.28$, and $\rho = 7900$ kg/m$^3$. The engine connecting rod is discretized into a grid model with 1045 nodes and calculated by three different numerical methods. Notably, the results of 78,526 nodes and 50,491 T-10 elements calculated by ANSYS [34,35] served as reference solutions.

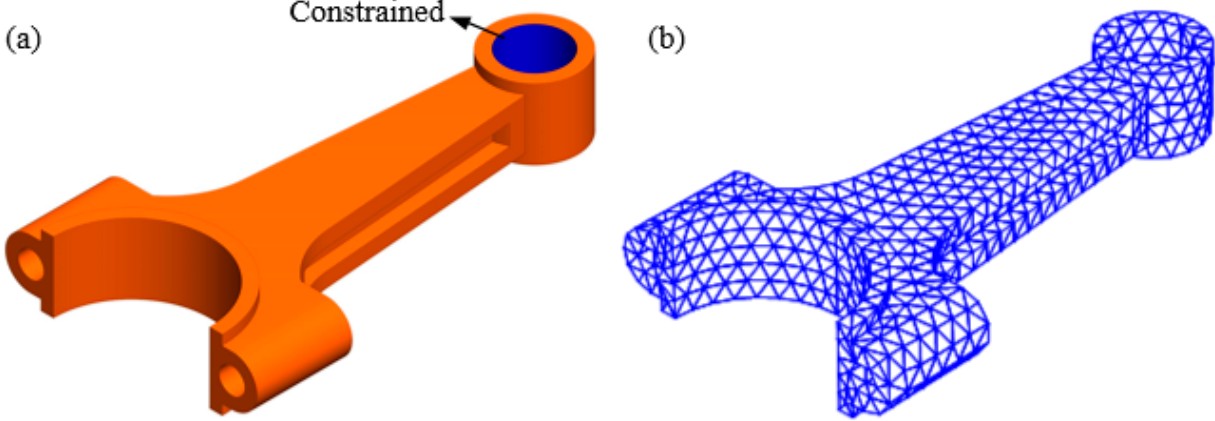

**Figure 24.** Connecting rod model: (**a**) geometric model; (**b**) mesh.

Table 10 shows the results of the first 15 natural frequencies [36,37]. In addition, FEM was used to calculate the fine mesh model with 5135 nodes; Figure 25 shows the relative errors of the first 15 free mode eigenvalues calculated by E-FEM, ES-FEM, and FEM. Based on the curves and data, the error of the results calculated by E-FEM was significantly smaller than that computed by ES-FEM and FEM, unlike the first 15 modal eigenvalues computed. Therefore, algorithm correctness in dealing with complex models can be preliminarily established. Figure 26 shows the first eight modes.

**Table 10.** First 15 modal eigenvalues of connecting rod.

| Order | FEM | ES-FEM | E-FEM | FEM (Fine) | Reference |
|---|---|---|---|---|---|
| $f_1$ | 582.14 | 528.66 | 525.51 | 541.21 | 522.99 |
| $f_2$ | 620.30 | 557.66 | 552.29 | 570.38 | 551.98 |
| $f_3$ | 1420.51 | 1105.39 | 1071.56 | 1160.39 | 1053.10 |
| $f_4$ | 3848.07 | 3571.62 | 3539.52 | 3624.49 | 3509.70 |
| $f_5$ | 4650.39 | 4152.56 | 4121.00 | 4256.49 | 4099.30 |
| $f_6$ | 5092.29 | 4688.78 | 4599.24 | 4751.62 | 4527.90 |
| $f_7$ | 8137.86 | 7794.29 | 7744.57 | 7881.27 | 7715.50 |
| $f_8$ | 10,191.39 | 8462.14 | 8271.87 | 8877.15 | 8104.20 |
| $f_9$ | 10,948.29 | 9737.92 | 9521.96 | 10,126.97 | 9242.10 |
| $f_{10}$ | 11,277.62 | 9977.75 | 9800.76 | 10,134.92 | 9624.20 |
| $f_{11}$ | 12,190.94 | 10,549.44 | 10,423.65 | 10,812.64 | 10,191.00 |
| $f_{12}$ | 15,146.03 | 12,015.30 | 11,664.54 | 12,570.43 | 11,327.00 |
| $f_{13}$ | 17,553.76 | 15,962.76 | 15,766.72 | 16,389.37 | 15,587.00 |
| $f_{14}$ | 19,701.50 | 17,353.71 | 17,125.72 | 17,825.40 | 16,870.00 |
| $f_{15}$ | 21,737.13 | 18,887.72 | 18,588.77 | 19,513.86 | 18,040.00 |

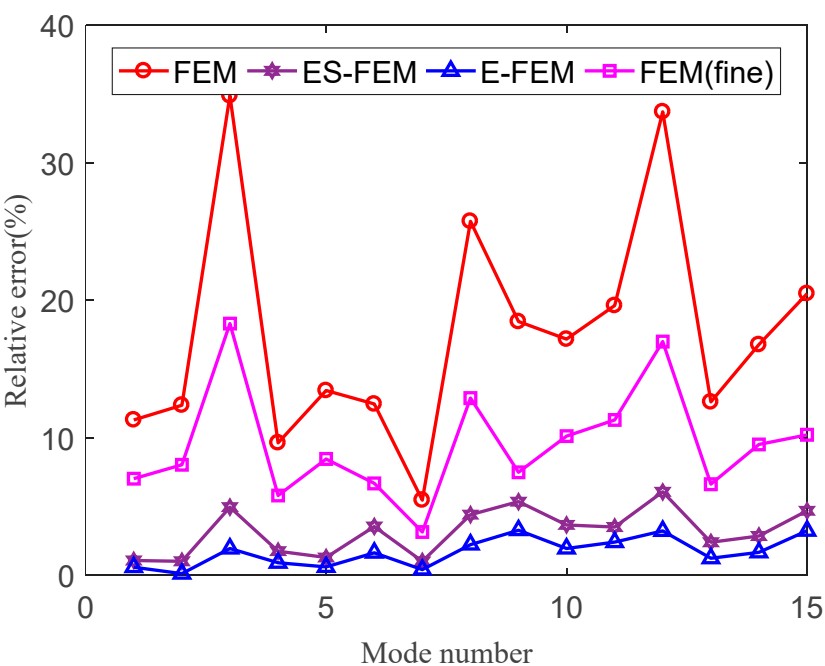

**Figure 25.** Relative error of first 15 modal eigenvalues of connecting rod.

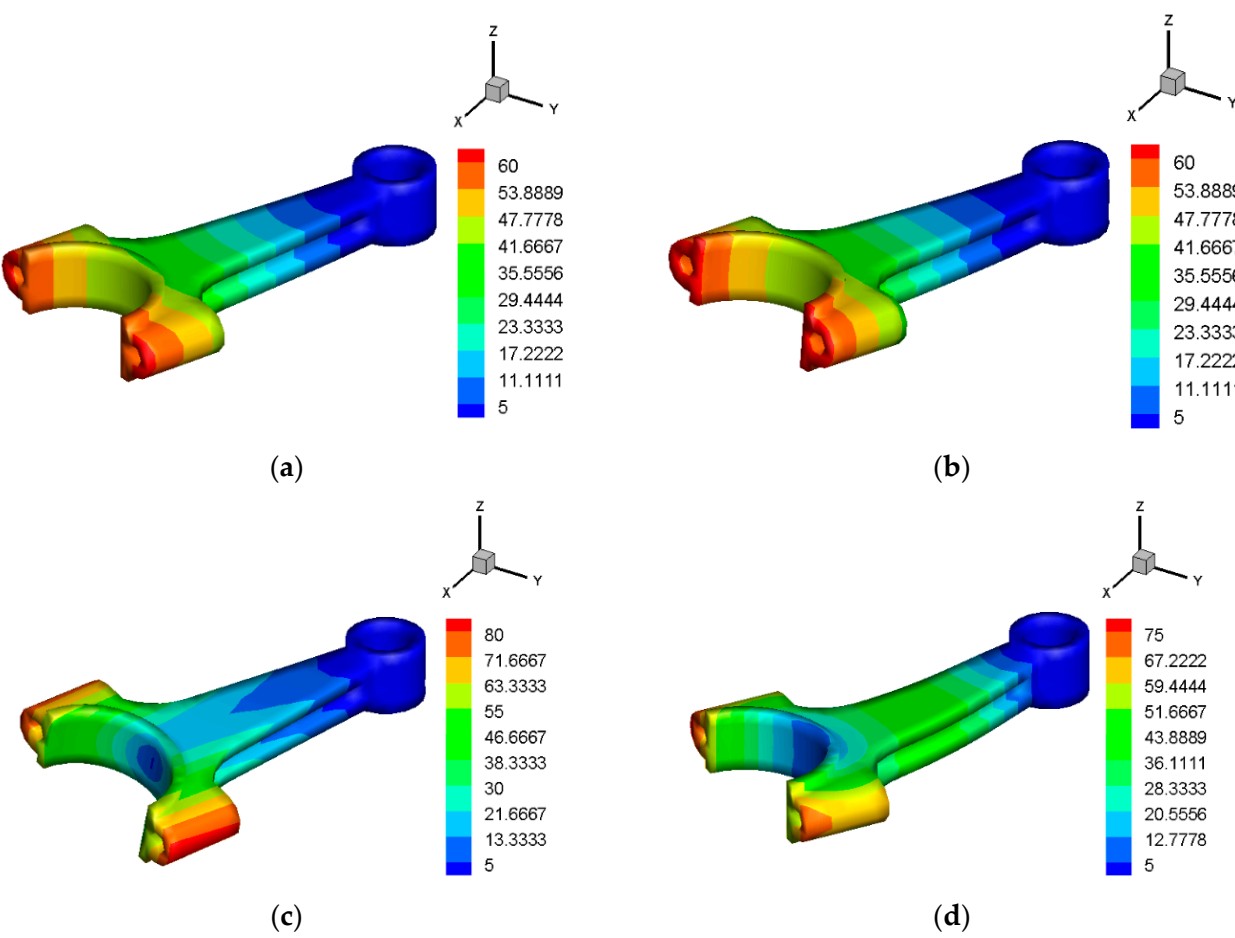

**Figure 26.** *Cont*.

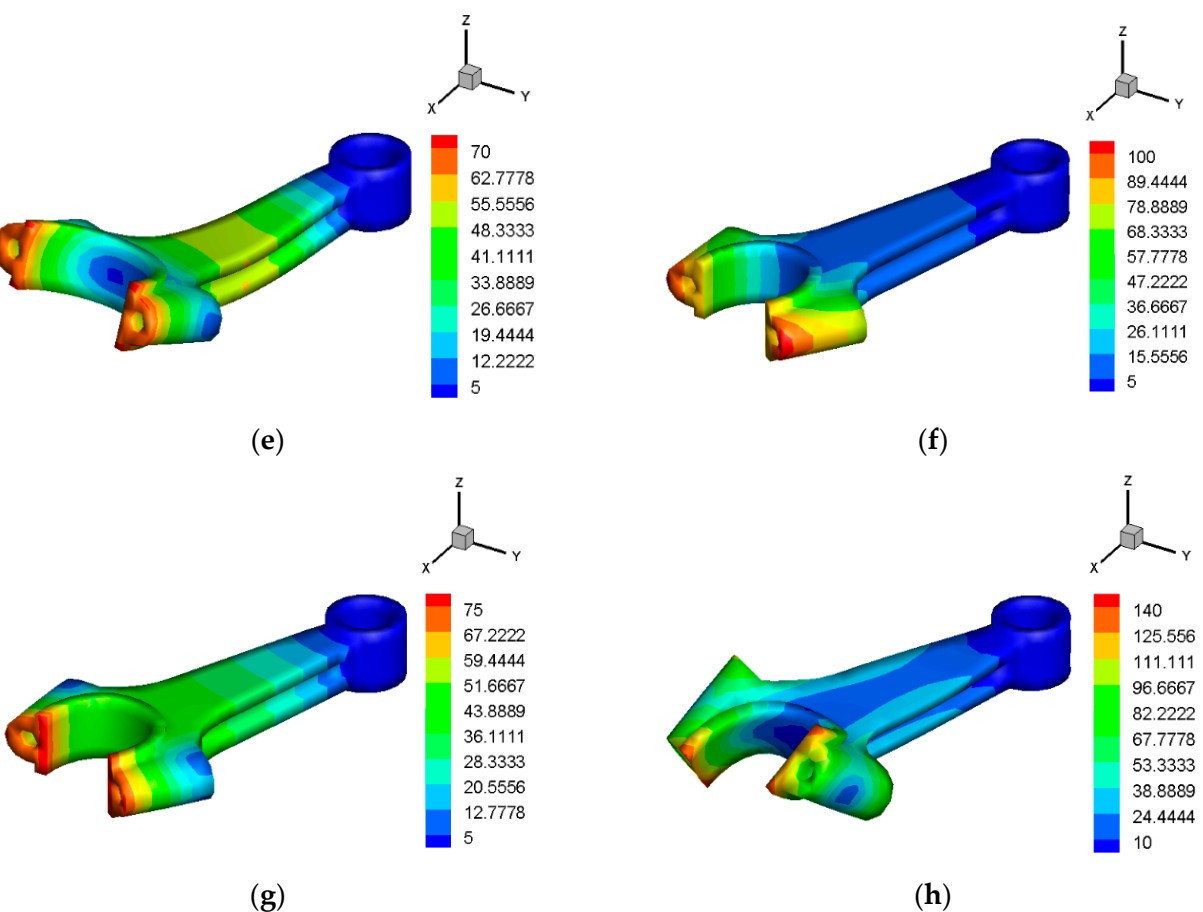

**Figure 26.** First 8 modal shapes of connecting rod calculated by E-FEM: (**a**) Mode one; (**b**) Mode two; (**c**) Mode three; (**d**) Mode four; (**e**) Mode five; (**f**) Mode six; (**g**) Mode seven; (**h**) Mode eight.

### 4.3. Automobile Front Suspension Arm

This section explores the applicability of the proposed method in calculating the complex geometric model of the front suspension arm. The geometric model of the front cantilever is shown in Figure 27, with fixed constraints on the blue surface. The material parameters are E = 69 GPa, $v = 0.3$, $\rho = 2700$ kg/m$^3$. The results of 54,087 nodes and 34,791 T-10 elements calculated by ANSYS were used as reference solutions.

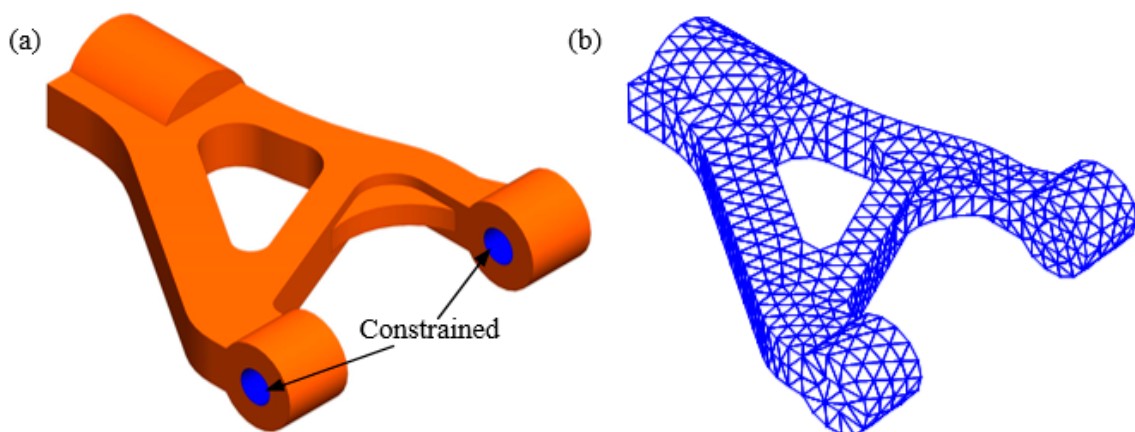

**Figure 27.** Vehicle suspension arm model: (**a**) geometric model (**b**) mesh.

In addition, FEM was used to calculate the fine mesh model with 5497 nodes, and the outcomes were compared using the three numerical approaches. Table 11 shows the results of the natural frequencies of the first 15 orders. In addition, the relative errors of free mode eigenvalues of the first 15 orders calculated by ES-FEM, FEM, and E-FEM are presented in Figure 28. According to the curves and data, the error of the results calculated by E-FEM was remarkably smaller than that computed by ES-FEM and FEM compared to the first 15 modal eigenvalues computed. Therefore, the reliability of the algorithm in dealing with complex models can be established. Figure 29 shows the first eight modes.

**Table 11.** Eigenvalues of the first 15 modes of the suspension arm.

| Order | FEM | ES-FEM | E-FEM | FEM (Fine) | Reference |
|---|---|---|---|---|---|
| $f_1$ | 193.75 | 159.44 | 157.18 | 167.46 | 155.32 |
| $f_2$ | 993.98 | 881.23 | 862.16 | 914.48 | 848.42 |
| $f_3$ | 1089.25 | 952.41 | 945.81 | 960.43 | 937.93 |
| $f_4$ | 1188.08 | 978.71 | 962.14 | 1018.88 | 950.71 |
| $f_5$ | 1877.99 | 1717.52 | 1703.89 | 1748.93 | 1686.60 |
| $f_6$ | 2134.42 | 2088.82 | 2080.74 | 2091.49 | 2063.20 |
| $f_7$ | 2675.84 | 2217.36 | 2163.85 | 2301.22 | 2130.50 |
| $f_8$ | 2792.76 | 2342.92 | 2291.51 | 2419.86 | 2256.90 |
| $f_9$ | 3279.08 | 2935.65 | 2906.53 | 3001.27 | 2863.10 |
| $f_{10}$ | 3790.99 | 3206.87 | 3105.93 | 3306.54 | 3055.20 |
| $f_{11}$ | 3857.71 | 3557.73 | 3526.29 | 3610.49 | 3478.20 |
| $f_{12}$ | 5209.41 | 4433.95 | 4348.32 | 4554.320 | 4289.50 |
| $f_{13}$ | 5422.11 | 4927.43 | 4863.51 | 5013.80 | 4734.90 |
| $f_{14}$ | 6159.29 | 5218.63 | 4981.44 | 5324.24 | 4878.10 |
| $f_{15}$ | 6637.36 | 5724.11 | 5529.93 | 5848.54 | 5451.40 |

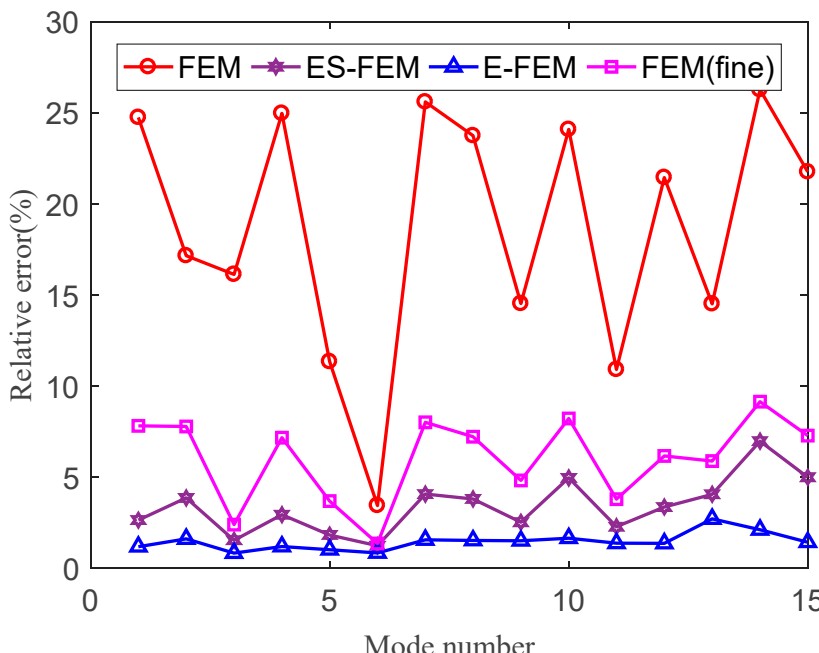

**Figure 28.** Relative error of first 15 modal eigenvalues of the suspension arm.

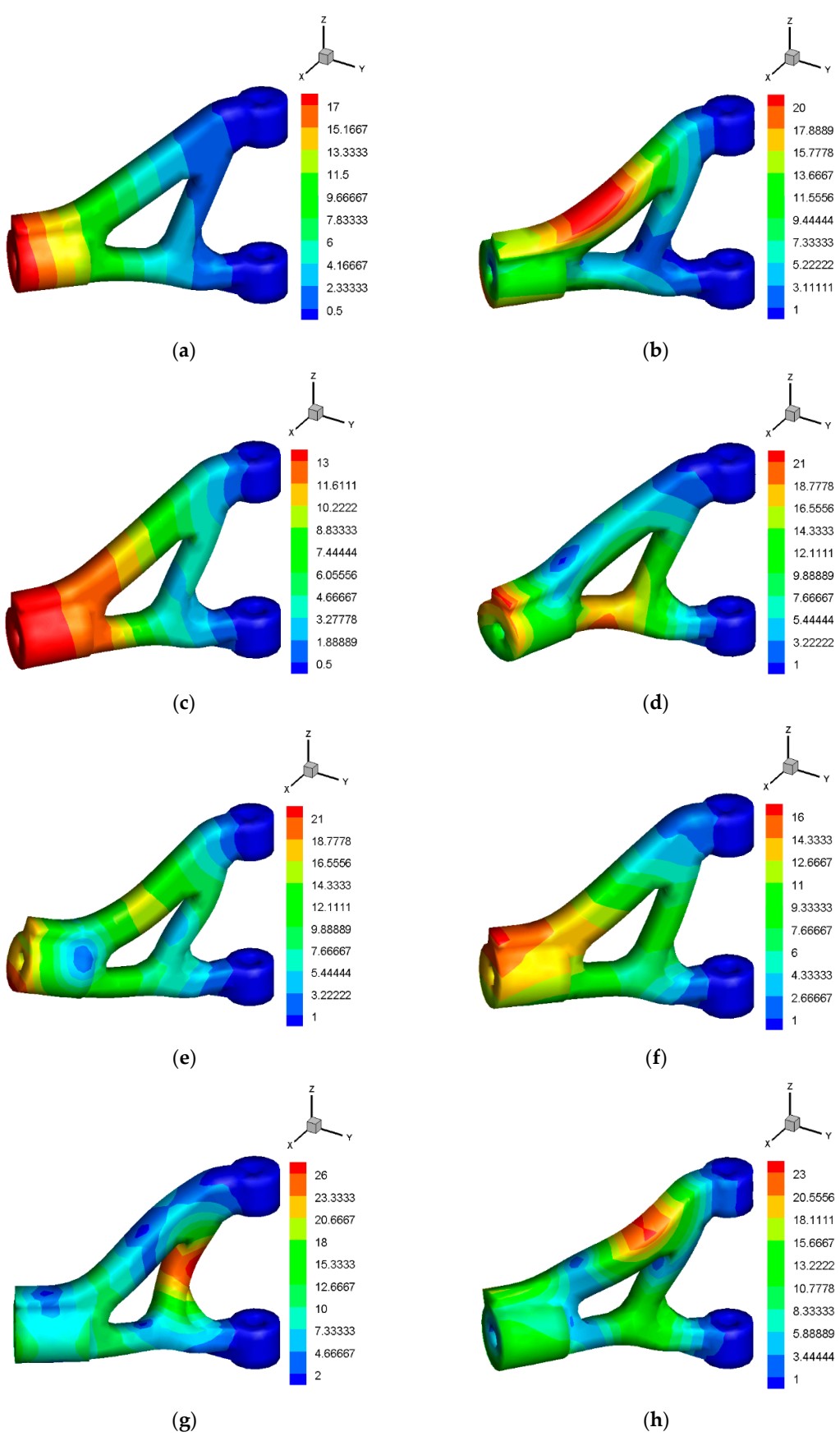

**Figure 29.** First 8 modes of automobile suspension arm calculated by E-FEM: (**a**) Mode one;(**b**) Mode two; (**c**) Mode three; (**d**) Mode four; (**e**) Mode five; (**f**) Mode six; (**g**) Mode seven; (**h**) Mode eight.

## 5. Conclusions

Based on the traditional FEM, this paper introduced a new basis function and proposed an improved E-FEM. The introduction of new shape functions, geometry matrices, stiffness matrices, and mass matrices improves variable displacement in the triangle and tetrahedral elements of classic FE. The considered numerical experiments can provide the following valuable insights: (1) The technique is used in 2D and 3D free vibration modal analysis, as well as comprehensively compares and analyzes computational efficiency, accuracy, and mesh distortion adaptability. E-FEM-T3 displays high accuracy, good convergence, and significant adaptability to grid distortion with four examples, i.e., the cantilever beam, connecting rod, shear wall, and automobile front suspension arm, under the presumption of not increasing too much calculation. (2) In the forced vibration study, E-FEM-T3, ES-FEM-T3, FEM-Q4, and FEM-T3 harbor extremely consistent vibration periods, whereas E-FEM-T3N6 has a highest amplitude, followed by E-FEM-T3N4, E-FEM-T3N3, ES-FEM-T3, FEM-Q4, and FEM-T3. The E-FEM-T3 has higher calculation precision and better adaptability to grid distortion by comparing computational efficiency to mesh distortion of the above methods in three different 2D models. (3) The proposed E-FEM-T3 element is also less sensitive to mesh distortion than other traditional elements, resulting in reliable numerical results, even when highly distorted models are used. Mesh distortion cannot always be avoided in engineering applications with complicated geometry; therefore, the current method is particularly well suited to these scenarios. (4) Although the current E-FEM-T3 requires no additional nodes compared to the high-order elements, it is more expensive numerically since larger system matrices are formed with more unknown nodal unknowns. Nevertheless, the current E-FEM-T3 still has better calculation precision and efficiency for free and forced vibration analysis solutions. (5) As the number of additional degrees of freedom increases, the convergence rate of the error relative to the calculation time for the three types of E-FEM-T3 solutions decreases; the higher the accuracy, the better the convergence, and the greater the adaptability to the distorted meshes can also be obtained.

**Author Contributions:** Methodology, Software, Q.G.; Writing—Original draft preparation, H.H.; Project administration, G.Z.; Funding acquisition, G.Z.; Methodology, F.W.; Software, F.W.; Supervision, Conceptualization, writing—review and editing, Z.J.; Visualization, Investigation, M.H.; Supervision, D.C.; Methodology, Data curation, Y.H. All authors have read and agreed to the published version of the manuscript.

**Funding:** This research was funded by the Science and Technology Support Program Project of Guizhou Province (Grant No. [2021] General 341), and funded by the Open Project of Key Laboratory of Architectural Acoustic Environment of Anhui Higher Education Institutes (Grant No. AAE2021ZD02) and Key Laboratory of Aeroacoustics, AVIC Aerodynamics Research Institute (No. XFX20220204).

**Data Availability Statement:** No additional data are available.

**Conflicts of Interest:** The authors declare no conflict of interest.

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
