# Peer review of "Enriched Finite Element Method Based on Interpolation Covers for Structural Dynamics Analysis"

_machines, doi:10.3390/machines11060587_

Round 1
Reviewer 1 Report
The paper has introduced a new basis function and proposed an improved E-FEM. The new method is based on the traditional FEM. The authors have explained the different types of methods by indicating appropriate and sufficient number of references. The E-FEM formula for structural dynamics analysis is thoroughly introduced in Section 3 of the paper.
Numerical experiments were conducted with different types of objects. The the triangle and tetrahedral elements were studied. The conclusions formulated by the authors are supported by the results.
In my opinion, the article would be of interest to specialists dealing with the finite element method. The results obtained and the proposed method can be useful in conducting various scientific studies. From a practical point of view, most users use CAE software to conduct a finite element analysis, where a new computational method cannot normally be introduced. However, I believe that the method proposed can be useful.
Author Response
Response to Reviewer 1 Comments
Point 1: The paper has introduced a new basis function and proposed an improved E-FEM. The new method is based on the traditional FEM. The authors have explained the different types of methods by indicating appropriate and sufficient number of references. The E-FEM formula for structural dynamics analysis is thoroughly introduced in Section 3 of the paper.
Numerical experiments were conducted with different types of objects. The the triangle and tetrahedral elements were studied. The conclusions formulated by the authors are supported by the results.
In my opinion, the article would be of interest to specialists dealing with the finite element method. The results obtained and the proposed method can be useful in conducting various scientific studies. From a practical point of view, most users use CAE software to conduct a finite element analysis, where a new computational method cannot normally be introduced. However, I believe that the method proposed can be useful.
Response 1:
Thanks for reviewer’s kindly comments. In this manuscript, we proposes a novel enriched finite element method (E-FEM) for structural dynamics analysis. We developed both the enriched 3-node triangular and 4-node tetrahedral displacement-based elements (T-elements). The forced and free vibration analyses were performed on various typical numerical examples. The proposed enriched finite element method generated more precise numerical results and ensured faster convergence than the original linear elements. Hopefully, through our research on numerical methods, we can provide some reference for the future development of CAE software technology.

Reviewer 2 Report
The manuscript is well written and organized. I do, however, have the following comments:
1. Line 15: "Higher-order obtained higher computational performance". Please define "higher computational performance". Faster computational speed or more accurate result?
Editorial:
1. Line 228: "ac curacy" -> "accuracy"
2. Caption of Figure 4: Should be (a) (b) (c) (d) instead of (a) (b) (a) (b).
Reviewer 3 Report
This paper describes a new finite element method called enriched (E-FEM). The approach improves 3-node and 4-node elements using interpolation functions. This improves the accuracy of the method while not adding to much additional computational time.
Forced and free vibrational analysis on a number of 2-D and 3-D examples show the capabilities of the method.
Even though individual steps are more computationally expensive, the method provides more precise numerical results and leads to faster convergence which offsets the increased computational cost of the individual step calculations.
Please add a table with the computational times for the steps and give the computer architecture used. Also give the overall computational times for the different approaches.
Please give literature references to the reference column where they seem to be missing in a number of tables.
They have different approaches for the E-FEM and show which ones work best in terms of improving the adaptability of the methods to different levels of grid distortion. It works better than other approaches to high levels of grid distortion where other methods will fail.
Overall, this is a nice paper with many details in terms of the method and a substantial number of examples. They also provide a large amount of ata in figures and tables which can be used to test other approaches.
Reviewer 4 Report
Comments in attachment
